# Interleaved resonance decays and electroweak radiation in the Vincia parton shower

**Helen Brooks[1], Peter Skands[2⋆] and Rob Verheyen[3†]**

**1** CCFE, Oxfordshire, UK
**2** School of Physics & Astronomy, Monash University, Clayton VIC-3800, Australia
**3** University College London, WC1E 6BT London, United Kingdom

⋆ peter.skands@monash.edu, † r.verheyen@ucl.ac.uk

## Abstract

We propose a framework for high-energy interactions in which resonance decays and electroweak branching processes are interleaved with the QCD evolution in a single common sequence of decreasing resolution scales. The interleaved treatment of resonance decays allows for a new treatment of finite-width effects in parton showers. At scales above their offshellness (*i.e.*, typically $Q > \Gamma$), resonances participate explicitly as incoming and outgoing states in branching processes, while they are effectively "integrated out" of the description at lower scales. We implement this formalism, together with a full set of antenna functions for branching processes involving electroweak ($W/Z/H$) bosons in the Vincia shower module in Pythia 8.3, and study some of the consequences.

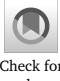

# 1 Introduction

In an interleaved evolution algorithm [1], two or more distinct types of shower-style resummations are performed jointly, by ordering them in a common measure of resolution scale and combining them into a single "interleaved" sequence of evolution steps. Examples are the interleaving of initial-state radiation (ISR) and multiple parton interactions (MPI) in Pythia 6 [1,2], the subsequent interleaving of final-state radiation (FSR) as well [3] in Pythia 8 [4], and more recently the interleaving of Pythia's MPI treatment with alternative FSR+ISR shower models such as Vincia [5] and Dire [6].

In such a framework, the full perturbative evolution probability can be written schematically as:

$$
\frac{d\mathcal{P}}{dQ^2} = \left( \sum_{\text{QCD}} \frac{d\mathcal{P}_{\text{QCD}}^{\text{ISR+FSR+MPI}}}{dQ^2} + \sum_{\text{QED}} \frac{d\mathcal{P}_{\text{QED}}^{\text{ISR+FSR}}}{dQ^2} \right)
$$
$$
\times \exp\left( -\int_{Q^2}^{Q_{i-1}^2} dQ'^2 \sum_{\text{QCD}} \frac{d\mathcal{P}_{\text{QCD}}^{\text{ISR+FSR+MPI}}}{dQ'^2} + \sum_{\text{QED}} \frac{d\mathcal{P}_{\text{QED}}^{\text{ISR+FSR}}}{dQ'^2} \right), \quad (1)
$$

as a function of a generic resolution variable, $Q$, for which both Pythia and Vincia use measures of transverse momentum, and where the sums run over (systems of) evolving QCD and QED charges respectively. The first line contains the unresummed probability densities (here

including MPI as well as the QCD and QED branching kernels) and the second line represents the combined Sudakov-like factor that expresses the total no-evolution probability between a preceding scale $Q_{i-1}$ and the current scale $Q$.

Interleaving provides a physically intuitive picture of the fine-graining of event structure with resolution scale. Moreover, to the extent that the different evolution steps can affect each other, e.g. by changing the available phase space and/or the number or character of the evolving entities, the joint resummation will have different properties than what a sequential or factorised approach would produce[1]. While we are not aware of a mathematical proof that interleaved evolution is more accurate than alternative approaches[2], we believe it is a physically well-motivated paradigm which is worth extending to further physics aspects.

One such aspect which, to our knowledge, has not been considered in an interleaved context until now is that of decays of short-lived resonances (though a formalism with similar effective consequences was put forth by Khoze and Sjöstrand in [9]). In hard processes that involve such resonances, the standard approach in Monte Carlo (MC) event generators is to adopt a factorised approach based on the narrow-width limit. Modulo spin-correlation effects, this is a reasonable starting point, in particular for resonances in the Standard Model (SM), which all have $\Gamma \lesssim \mathcal{O}(\text{GeV}) \ll M$. Thus, as a first step, one performs ISR+FSR on the Born-level hard process treating any outgoing resonances as stable. To account for finite-width effects, one could in principle impose a cutoff or dampening of radiation off resonances at scales of order their widths, to reflect that they cannot physically radiate coherently at wavelengths longer than $\sim \hbar c / \Gamma$, but since this is not a big effect for SM resonances (see e.g., [9–11]), such effects are currently not accounted for in Pythia. In a second step, one iteratively treats each resonance decay, adding additional FSR showering to each decay process as needed. In these secondary showers care must be taken to preserve the invariant mass of each resonance-decay system, in order not to generate unphysical corrections to the Breit-Wigner shapes. Again, perturbative finite-width effects are typically regarded as small [9–11] and are not formally included in Pythia's current modelling.

In section 2, we extend the concept of interleaving to include decays of short-lived resonances. This opens for a new treatment of finite-width effects, in which resonances can act as physical emitters (and recoilers) during an initial stage of the evolution but effectively are "integrated out" of the physics description at scales below $Q \sim \Gamma$. This automatically suppresses low-frequency radiation off the resonances and allows for low-scale interference effects to generate modifications to the Breit-Wigner resonance shapes. The model has qualitative similarities with (but is not identical to) the preferred scenario of [9]. Moreover we propose a recursive picture which allows for a natural treatment of sequential (cascade) decays.

A related aspect concerns how weak-scale resonances are created in the first place and, more generally, how electroweak (EW) corrections are treated. Pure QED showers aside, the dominant approach is to compute cross sections using fixed-order expansions in the relevant weak couplings. This is a reasonable starting point at relatively low ("electroweak-scale") energies where the integrated probabilities for multiple weak branching processes over phase space, and hence the corresponding electroweak logarithms (see, e.g., [12]), are small. An alternative approach which is better suited for (asymptotically) high energies is that of electroweak showers [13–17], by which electroweak logarithms can be resummed to all orders, at the expense of neglecting subleading- and process-dependent non-logarithmic terms in a similar compromise as that made in ordinary (QCD/QED) showers[3]. The current implementation of EW showers in Pythia [13] focuses on spin-averaged vector-boson ($V \in [W, Z]$) emissions in jets only. Notably, it does not include triple-boson vertices such as $V \to VV$, $V \to VH$, etc.,

---

[1]See, e.g., [7] for a recent analytical study which at least in part goes in this direction.

[2]There are, however, cases for which non-interleaved evolution can be shown to give wrong answers [8].

[3]We note that merging techniques for EW showers have been developed to address this [18].

nor does it account for the spin dependence that arises due to the chiral nature of the EW sector. Three alternative comprehensive formalisms have since been developed, in [15–17].

In sec. 3, we adapt the EW shower formalism of [16] to the $p_\perp$-ordered Vincia shower model and discuss its implementation in terms of antenna functions, evolution variables, Sudakov factors, recoil strategies, and treatment of neutral-boson interference effects. The implementation includes a full set of explicit EW antenna functions with helicity dependence, that are recapitulated in app. A, with overestimating trial integrals for $p_\perp$-evolution collected in app. B. The combination of a full-fledged EW shower with other parts of MC event generation raises a number of issues. In sec. 3.5, we discuss how we treat branching processes that are present both in the EW shower and as resonance decays, such as $t \to bW$, $Z \to q\bar{q}$, etc. Furthermore, in sec. 3.6 we discuss how to avoid double-counting between EW and QCD showers from different Born-level hard processes, such as, e.g., in the case of $pp \to VVj$ which can be reached both via a Born-level $VV$ event + a QCD emission, or via a Born-level $Vj$ event + an EW $V$ emission.

Extending eq. (1) to include interleaved resonance decays and EW shower branchings, it becomes:

$$
\frac{d\mathcal{P}}{dQ^2} = \left[\left(\sum_{\text{QCD}} \frac{d\mathcal{P}_{\text{QCD}}^{\text{ISR+FSR+MPI}}}{dQ^2} + \sum_{\text{EW}} \frac{d\mathcal{P}_{\text{EW}}^{\text{ISR+FSR}}}{dQ^2}\right)\left(1 - \sum_{\text{RES}} \int_{Q^2}^{Q_{i-1}^2} dQ'^2 \frac{d\mathcal{P}^{\text{RES}}}{dQ'^2}\right) + \sum_{\text{RES}} \frac{d\mathcal{P}^{\text{RES}}}{dQ^2}\right]
$$

$$
\times \exp\left[-\int_{Q^2}^{Q_{i-1}^2} dQ'^2 \left(+\sum_{\text{QCD}} \frac{d\mathcal{P}_{\text{QCD}}^{\text{ISR+FSR+MPI}}}{dQ'^2} + \sum_{\text{EW}} \frac{d\mathcal{P}_{\text{EW}}^{\text{ISR+FSR}}}{dQ'^2}\right)\right], \tag{2}
$$

where the sum over $d\mathcal{P}^{\text{RES}}/dQ^2$ in the second line (and the corresponding negative integral in the first line) expresses the interleaving of resonance decays with the rest of the evolution via probability densities for $1 \to N$ "branchings" (decays), each normalised to integrate to unity, and the full set of EW antenna kernels, $\mathcal{P}_{\text{EW}}$, has replaced the corresponding QED ones in the first term. The precise interpretation of eq. (2) will be elaborated on in secs. 2 – 3 below.

Finally, in sec. 4 we present a set of validations and preliminary results, and use them to discuss the physical implications of our approach, before summarising and concluding in sec. 5. The complete set of EW antenna functions is collected in app. A. Integrals of trial-antenna functions are worked out in app. B. Methods for Breit-Wigner sampling and expressions for the partial widths of Higgs bosons, transversely and longitudinally polarised vector bosons, and top quarks are collected in app. C.

## 2 Interleaved Resonance Decays

The starting point for the modelling of resonance production and decays in event generators is the narrow-width limit, $\Gamma/M \to 0$, also called the pole approximation. In this limit, an infinite timelike interval separates the production and decay of the resonance, and there is no interference between radiation emitted before and after the decay. Formally, the decay of a narrow resonance can be *factorised* from its production process.

If the resonance carries spin, the factorisation takes the form of a tensor in spin space [19, 20]. This can be incorporated in Monte Carlo simulations through a clever linearly scaling algorithm [21–25], as is done for instance in Herwig [24, 26], while Pythia employs spin-averaged expressions with correlations imposed on a case by case basis e.g. by using the full 4-fermion matrix element to generate correlated decay angles for the two $W$ bosons in $e^+e^- \to W^+W^- \to 4$ fermions. In both cases, spin correlations (when they are included)

manifest themselves as angular correlations between partons from different decays and/or between partons from a decay and partons in the hard process.

In addition, event generators typically make several *finite-width* improvements on the strict narrow-width assumption, notably Breit-Wigner distributions $BW_R(Q^2)$ for resonance invariant masses instead of $\delta$ functions, as well as options for allowing partial widths (and hence relative branching fractions) to vary with $Q^2$. The latter allows, for example, to account for kinematic thresholds and effects of running couplings across a reasonable range around the pole, but cannot be pushed too far, especially in the electroweak sector where masses and couplings are not independent of each other.

However, the production and decay of the resonance is still treated separately, without accounting for any (perturbative) interference effects between them beyond colour conservation and, in some cases, spin correlations. We refer to such treatments as Breit-Wigner-improved pole approximations (BWPA).

For example, in both Pythia and Herwig, a hard process like $gg \to t\bar{t}$ (with independently selected Breit-Wigner distributed masses for both tops) is first subjected to both initial- and final-state showers starting at the evolution-scale maximum defined by the hard process and ending at the infrared shower cutoff. After this, each of the top-decay processes, $t \to bW$, are subjected to an internal "resonance shower". The latter is done in a way that preserves the invariant mass of the resonance-decay system so that the Breit-Wigner shape of the decaying top quark is preserved (*i.e.*, there are no momentum exchanges with any partons outside of the top-decay system), again only stopping when the infrared shower cutoff is reached. Finally the $W$ decay systems are showered similarly.

The implicit assumption is that interference between radiation emitted in each of these stages (top production, top decay, and $W$ decay) is negligible. The fundamental reason why this is a good assumption, at least for perturbative QCD radiation off SM particles, is that none of the SM resonances (top, Higgs, $W$, and $Z$ bosons) have widths that are much larger than the shower cutoff for QCD radiation, $Q_{\text{cut}} \sim 1\,\text{GeV}$, hence the region of the phase space for perturbative QCD shower evolution over which interference effects could be relevant is very small. The strong suppression of such interference effects have also been verified by explicit theoretical and phenomenological studies e.g. of $e^+e^- \to W^+W^-$ [27] and $e^+e^- \to t\bar{t}$ [9,10, 28,29].

Nevertheless, the experimentally achievable statistical precision on top-quark mass measurements at hadron colliders has now reached the order of a few hundred MeV [30–33], making it important to evaluate (and preferably control) QCD uncertainties at that level or better. This has catalysed a reassessment of possible non-perturbative uncertainties such as colour reconnections [34–36], and also of the effects of soft perturbative radiation [37, 38] and of finite-width effects in fixed-order matrix elements matched to parton showers [39,40]. So far, the latter efforts have focused mainly on improvements to the treatment of finite-width effects on the fixed-order side, and on how to match these consistently with showers, without substantial modifications to the showers themselves.

Here, we note that the BWPA is, strictly speaking, not quite consistent with the strong-ordering condition in parton showers. Strong ordering expresses the simple fact that the leading singularity structures of QCD (and QED) amplitudes correspond to Feynman diagrams in which each successive propagator has a much smaller virtuality than the preceding one (or next one, for initial-state legs). Physically, this reflects a formation-time principle; short-lived fluctuations do not have time to emit low-frequency radiation. However, for unstable particles in the BWPA, one can have precisely the situation that a particle which has nominally been assigned an invariant mass quite different from the pole value does emit low-frequency radiation. In the corresponding Feynman amplitudes, there are then two (or more) off-shell propagators, which ought to be suppressed relative to amplitudes in which the low-frequency

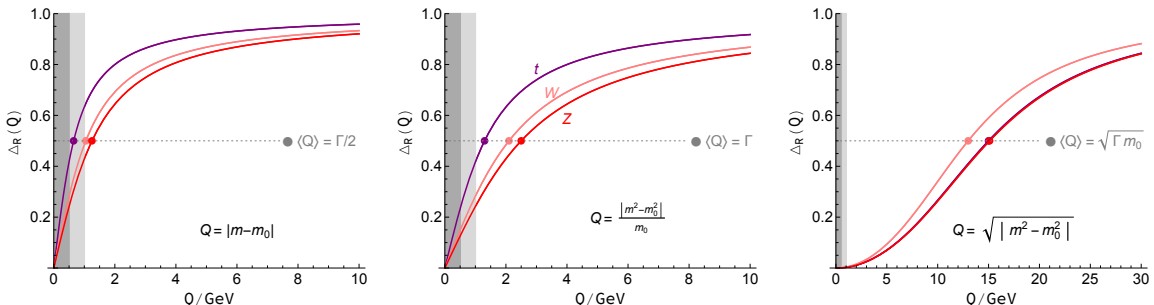

Figure 1: Resonance survival fractions $\Delta_R$, as functions of evolution scale $Q$, for $Q_{\text{RES}} = |m - m_0|$ (left), $Q_{\text{RES}} = |m^2 - m_0^2|/m_0$ (middle), and $Q_{\text{RES}} = \sqrt{|m^2 - m_0^2|}$ (right). Each plot shows the surviving fraction of top quarks, $W$ bosons, and $Z$ bosons, in purple, pink, and red, respectively. A typical range of QCD shower cutoff values, from 0.5 to 1 GeV, is shown with light gray shading, while the region below 0.5 GeV is shaded darker gray. (As of Pythia 8.306, the default value in Vincia is 0.75 GeV, in the middle of the light-gray region.) The median values of $Q_{\text{RES}}$, corresponding to $\Delta_R = 0.5$, are highlighted by dots on each curve.

radiation is emitted after the decay. This leads us to consider an interleaved paradigm for showers off resonance-production + decay processes, in which resonance decays are inserted in the overall event evolution when the perturbative evolution scale reaches a value of order the width of the resonance.

The desire to connect with the strong-ordering criterion in the rest of the perturbative evolution, as the principle that should dictate the leading amplitude structures, leads us to prefer a dynamical scale choice for resonance decays, whereby resonances that are highly off shell will persist over shorter intervals in the evolution than ones that are almost on shell. We note that this has the consequence that the on-shell tail will be resolvable by soft photons or gluons, albeit suppressed by the survival fraction. To illustrate this, fig. 1 shows the survivial fractions (denoted $\Delta_R$) as functions of evolution scale, for $t$, $Z$, and $W$ resonances, for three different options for dynamical scale choices, all of which are roughly motivated by the propagator structure:

$$i) \quad Q_{\text{RES}}^2 \quad = \quad (m - m_0)^2 \,, \tag{3}$$

$$ii) \quad Q_{\text{RES}}^2 \quad \overset{\text{default}}{\equiv} \quad \left( \frac{m^2 - m_0^2}{m_0} \right)^2 > 0 \,, \tag{4}$$

$$iii) \quad Q_{\text{RES}}^2 \quad \equiv \quad |m^2 - m_0^2| \,, \tag{5}$$

where $m_0$ is the pole mass and $m$ its BW-distributed counterpart. Near resonance, options *i)* and *ii)*, illustrated in the left and middle panes of fig. 1, are functionally almost equivalent, differing mainly just by an overall factor 2, while for option *iii)*, illustrated in the rightmost pane, $m = m_0 \pm \Gamma/2$ translates to $Q^2 \sim m_0 \Gamma$, so that option is primarily intended to give an upper bound on the effect that interleaving could have.

Alternatively, our model also allows for using a fixed scale, $Q_{\text{RES}} \equiv \Gamma$, irrespective of off-shellness. In that case, the resonance will not be resolved at all by any photons or gluons with scales $Q < \Gamma$. We regard this as a good starting point for the width dependence but have not selected it as our default since the fixed-scale choice by itself does not automatically extend strong ordering to the resonance propagators; this can only be achieved by allowing the choice to be dynamical. Our default choice, eq. (4), is constructed to have a *median* scale of $\langle Q_{\text{RES}} \rangle = \Gamma$, while simultaneously respecting strong ordering event by event. This implies that

soft quanta will be able to resolve the resonance with a suppressed magnitude $\propto \Delta_R$, which acts as a form factor.

Finally, interleaved resonance decays, and especially the dynamical scale choices, have the additional benefit of enabling a particularly simple way to match resonance decays and EW branching processes. This is necessary since processes such as $t \to bW$ occur not only as resonance decays but also as EW shower branchings (and for sufficiently high energies can occur multiple times); from the point of view of the EW shower, the "decay" process is merely the last such branching, which happens with unit probability. Since the EW shower evolution is ordered in a measure of propagator offshellness, a simple and elegant transition between the EW shower evolution which dominates at very high offshellnesses and the unit-probability Breit-Wigner-distributed resonance decays which dominate near the pole, can be achieved by also ordering the latter in offshellness, *viz.* by interleaved resonance decays. We return to this in sec. 3.5.

## 2.1 The Model

The change to an interleaved treatment of resonance decays is achieved by introducing a probability density for $1 \to n$ decay(s) in the perturbative evolution, as follows:

1. We assume that we start from a hard process (or a previous decay or EW shower branching process) which has produced a resonance according to the BWPA, *i.e.* a distribution of the form

$$\mathrm{BW_R}(m; m_0, \Gamma) = \frac{N}{\pi} \frac{m_0 \Gamma}{\left(m^2 - m_0^2\right)^2 + \left(m_0 \Gamma\right)^2} \,, \tag{6}$$

   where $m_0$ is the pole mass and $\Gamma$ is the width of the resonance[4]. The overall normalisation factor $N$ ensures that the probability density integrates to unity. In this work, the above formula is also used for resonances produced by the EW shower (see below), with explicit expressions given in app. C.

2. By default, we define the evolution scale associated with the decay of the resonance to be given by eq. (4). As discussed above, this implies, e.g., that the decay of a resonance which has $m = m_0 \pm \Gamma/2$ will be performed when the interleaved perturbative event evolution reaches a scale $Q \sim \Gamma$, *cf.* the middle pane of fig. 1. The left- and right-hand panes illustrate two alternative dynamical choices, with lower and higher averages, respectively.

3. Having determined a scale $Q_{\mathrm{RES}}$ for each resonance in the hard process, the interlaved perturbative event evolution commences, at the shower/MPI starting scale, $Q_0$, determined by the hard process.

4. If any $Q_{\mathrm{RES}}$ values are greater than $Q_0$, the corresponding resonance decays are performed immediately; *i.e.*, those resonances are replaced by their decay products. Any other resonances are kept stable (for now).

5. The interleaved event evolution continues (with shower emissions and multi-parton interactions generated as usual), until either the shower cutoff or a $Q_{\mathrm{RES}}$ scale is reached.

When a $Q_{\mathrm{RES}}$ scale is reached, the following happens:

---

[4]We note that, in Pythia, both the total width and the branching fractions into individual channels, can in principle depend on $m$. This is, e.g., used to model threshold behaviours, and to ensure $\Gamma(m) \to 0$ when there are no channels kinematically open. See [2] for further discussion on this point.

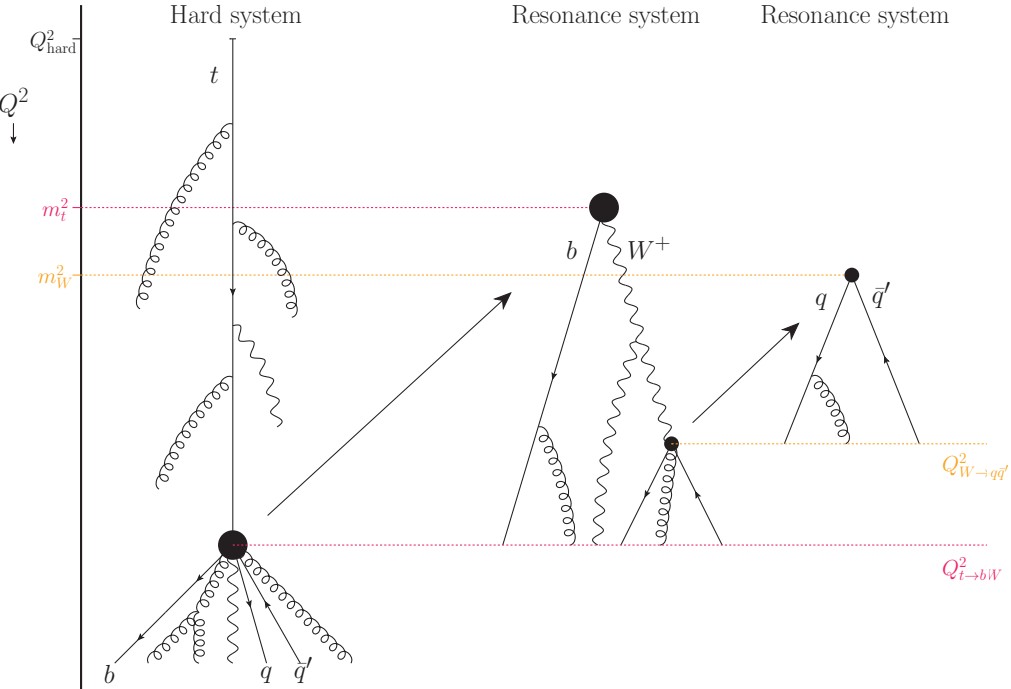

Figure 2: Recursive interleaving of resonance decays with QED and QCD radiation, for a hard event with $t \rightarrow bW$ followed by $W \rightarrow q\bar{q}'$. The vertical axis represents the evolution scale, from high (top) to low (bottom). The scales $m_t$ and $m_W$ represent the (BW-distributed) invariant masses of the respective particles, setting upper kinematic limits for each resonance shower. The new aspect is the introduction of the scales $Q_{t \rightarrow bW}$ and $Q_{W \rightarrow q\bar{q}'}$ (typically of order $\Gamma_t$ and $\Gamma_W$ respectively), below which each of the resonance-decay systems are merged into their respective production system(s).

6a. The resonance in question is replaced by its decay products. (From the point of view of the rest of the perturbative evolution, this looks like a $1 \rightarrow n$ branching.) To give an example, the diagram in fig. 2 illustrates a hard process with an outgoing top quark. Initially, the top quark is treated as a stable outgoing particle which is subject to showering as usual, but when the evolution scale reaches the scale associated to the top-quark decay, here denoted $Q_{t \rightarrow bW}$, the evolution of the hard event is put on hold, and the top-decay treatment is started. This step is represented by the leftmost diagonal black arrow.

6b. A "resonance shower" begins. As described in detail for non-interleaved resonance showers in [38], this stage *only* involves the decaying resonance and its decay products, with the four-momentum and quantum numbers of the initial resonance preserved as boundary conditions. There are no momentum (or quantum number) transfers to/from any partons outside of the resonance-decay system. The starting scale $Q'_0$ for the resonance shower is set to the phase-space maximum for FSR in the given decay process, exactly as in the non-interleaved case. In fig. 2, the change from $Q_{RES} = Q_{t \rightarrow bW}$ to $Q'_0 = m_t$ is illustrated by red-dotted lines.

6c. Any new resonances produced by the decay process are associated with their own $Q'_{RES}$ scales, which can be higher or lower than $Q_{RES}$. In the example of fig. 2, the top decay produces a $W$ boson, so a scale for the $W$ boson decay, $Q_{W \rightarrow q\bar{q}'}$, is determined.

6d. The resonance shower continues, with FSR shower branchings, until the evolution reaches either the next $Q'_{\text{RES}}$, or the original scale $Q_{\text{RES}}$. Fig. 2 exemplifies the former case, with $Q_{W \to q\bar{q}'} > Q_{t \to bW}$. (Since $\Gamma_W > \Gamma_t$ in the SM, this is the typical case; it also allows us to illustrate the recursive nature of our approach.) Thus, the scale associated to the $W$ decay, $Q_{W \to q\bar{q}'}$, will be reached before the end of the top-decay resonance-shower stage.

6e. When a nested $Q'_{\text{RES}}$ is reached, the current resonance shower is placed on hold, while the decay + shower of the corresponding nested resonance decay occurs, recursively, starting back from step 6a. In the example of fig. 2, this is illustrated by the rightmost diagonal black arrow and the subsequent shower evolution in the $W \to q\bar{q}'$ system.

6f. When the resonance-shower evolution again reaches $Q_{\text{RES}}$, the resonance-shower stage ends, and the resonance-decay system (including any radiation emitted as well as any nested systems treated so far) is merged with the parton system that produced it. This is illustrated in fig. 2 by the bottom edges of the two resonance-shower systems, at which the showered partons are collected into an effective vertex which is inserted in the parent process. The evolution of the merged system then recommences at step 5, starting from $Q_{\text{RES}}$. Momentum exchanges between partons that originated from the decayed resonance and partons from outside that system may now in principle occur, to the extent prescribed by colour and/or charge flow.

When $Q_{\text{cut}}$ is reached, the following happens:

7. As a last step, any resonances with $Q_{\text{RES}}$ below $Q_{\text{cut}}$ are decayed in no particular order, just as in the conventional (non-interleaved) treatment of resonance decays. However, any nested decays that have $Q'_{\text{RES}} > Q_{\text{cut}}$ will still be performed recursively, as above. For example, the decay of a top quark with $Q_{\text{RES}} < Q_{\text{cut}}$ will take place after the shower evolution of its production process has finished, but if the $W$ produced in its decay has $Q'_{\text{RES}} > Q_{\text{cut}}$, then that decay will still be interleaved within the top-decay shower evolution.

Extending eq. (1) to include interleaved resonance decays, it becomes:

$$
\begin{aligned}
\frac{\mathrm{d}\mathcal{P}}{\mathrm{d}Q^2} = {} & \left[ \frac{\mathrm{d}\mathcal{P}^{\text{RES}}}{\mathrm{d}Q^2} + \left( \frac{\mathrm{d}\mathcal{P}^{\text{MPI}}}{\mathrm{d}Q^2} + \frac{\mathrm{d}\mathcal{P}^{\text{ISR+FSR}}}{\mathrm{d}Q^2} \right) \left( 1 - \int_{Q^2}^{Q_{i-1}^2} \mathrm{d}Q'^2 \frac{\mathrm{d}\mathcal{P}^{\text{RES}}}{\mathrm{d}Q'^2} \right) \right] \\
& \times \exp \left[ - \int_{Q^2}^{Q_{i-1}^2} \mathrm{d}Q'^2 \left( \frac{\mathrm{d}\mathcal{P}^{\text{MPI}}}{\mathrm{d}Q'^2} + \frac{\mathrm{d}\mathcal{P}^{\text{ISR+FSR}}}{\mathrm{d}Q'^2} \right) \right],
\end{aligned}
\tag{7}
$$

where it is understood that the ISR+FSR term includes a sum over QED and QCD radiators, and similarly the RES term includes a sum over decayers.

Different from conventional interleaved parton-shower and MPI kernels, the term $\mathrm{d}\mathcal{P}^{\text{RES}}/\mathrm{d}Q^2$ is not exponentiated in the Sudakov factor. This is because the probability density expressed by the Breit-Wigner distribution is already unitary and contains its own infinite-order resummation. In other words: if a resonance is produced, its decay happens once, and once only; there is no need for a Sudakov-style resummation of it. Due to the interleaving with in particular the EW shower, there is, however, a finite probability (given by the EW Sudakov factor) that the resonance will undergo one or more EW branchings before it gets a chance to decay. We return to this in sec. 3.

We emphasise that, at the current stage, this proposal can only be considered a heuristically motivated paradigm. Applying the strong-ordering principle to finite-width propagators produces a kind of forced marriage between two different all-orders summations, the self-energy

Breit-Wigner one, and the LL bremsstrahlung one. It captures the basic feature that radiation at frequencies below the resonance width should be suppressed, and we therefore consider it of phenomenological interest to explore its consequences. Should it become relevant to the community, a more formal mathematical investigation would be welcome.

Note also that the systematic inclusion of non-resonant effects would require future extensions of matching strategies, beyond the scope of this paper to explore.

A final point left for possible future investigations is that resonances with low off-shellnesses can in principle persist to arbitrarily low scales. This raises the question whether, e.g., top quarks that are assigned off-shellness values less than the infrared shower cutoff (or less than $\Lambda_{\text{QCD}}$) should be allowed to hadronise.

## 2.2 Summary of Consequences

To summarise, the main consequences of the interleaving of resonance decays with the rest of the perturbative evolution are:

- Due to the interleaving, unstable resonances effectively disappear from the evolution at an average scale $Q \sim \Gamma$. They will therefore not be able to act as emitters or recoilers for radiation below that scale; only their decay products can do that.

- After the resonance has disappeared, recoils to partons originating outside of the decay system are in principle allowed, and may distort the Breit-Wigner shape. In practice, such recoil effects are still expected to be relatively small, for several reasons. Firstly, the fact that the interleaving only "kicks in" below the offshellness scale limits any out-of-resonance recoil effects (e.g., in terms of $p_\perp$ kicks) to be smaller than that scale. Secondly, in decays of QCD colour singlets, such as $Z$ and $W$ bosons, there are no leading-colour (LC) dipoles to any partons outside of the decay system and hence no out-of-resonance QCD recoils at all. Even top-quark decays only involve a single such connection, corresponding to the colour flowing through the decay. Analogous arguments also apply to QED radiation, with $\alpha_s \to \alpha_{\text{EM}}$ and the colour of the resonance replaced by its overall electric charge.

- With the dynamical choice of decay scale, highly off-shell particles disappear from the evolution at higher evolution scales than ones nearer the pole mass value, translating to an increasing distortion of the resonance shape further away from the pole. This roughly corresponds to the notion of strong ordering in the rest of the evolution.

At the technical level, the interleaved handling of resonance showers does not dramatically change the amount of time it takes to generate events. To illustrate this, we consider fully showered and hadronised $e^+e^- \to t\bar{t}$ events at $\sqrt{s} = 500$ GeV, taking the case of sequential decays with $\Gamma_t = 1.5$ GeV as reference. Fig. 3 shows the relative change in run time for the fixed-scale and default dynamical-scale choices, as functions of the top quark width, for the case of stable $W$ bosons (left plot) and with $W$ decays included (right plot). Note that there is an overall uncertainty on the time measurements of a few per-cent.

Without $W$ decays (left plot), the total run-time is nearly the same with and without interleaving for the SM value of the $\Gamma_t$, while interleaving actually decreases the run-time for large widths. We interpret this as due to the non-interleaved treatment having to perform two evolutions in the range $\Gamma_t > Q > Q_{\text{cut}}$, while the interleaved treatment only goes through that region once, removing the double-counting. In the right plot, with $W$ decays included, the interleaved treatment is 10% - 20% slower than the sequential one. (Note: the $W$ width was not varied, only the top-quark one.) For $pp \to t\bar{t}$, interleaving does not measurably change the total run time at all, which is dominated by the generation of multi-parton interactions, colour reconnections, and hadronization.

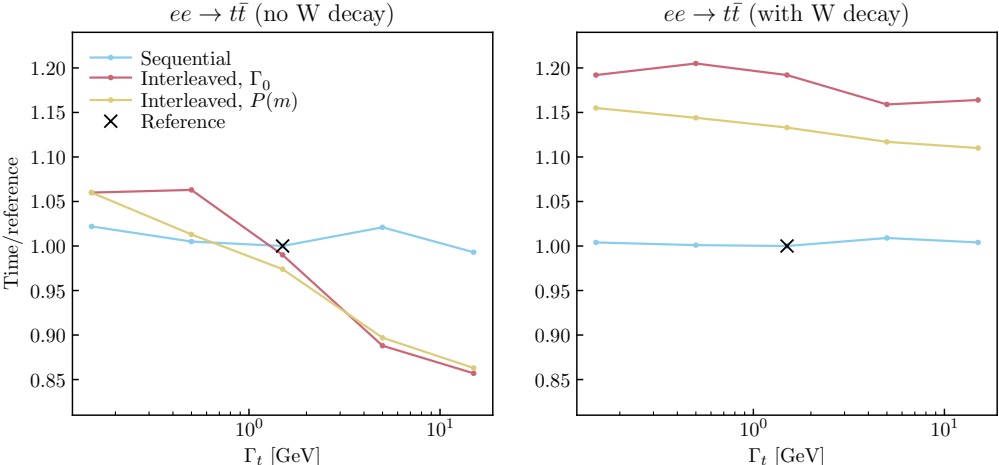

Figure 3: Relative time consumption to generate $e^+e^- \to t\bar{t}$ events at $\sqrt{s} = 500\,\text{GeV}$, as functions of the top quark width, for sequential (blue) and interleaved resonance decays with fixed (red) and the default dynamical (yellow) scale choices. *Left:* with stable *W* bosons. *Right:* with inclusively decaying *W* bosons. In both cases, sequential decays with $\Gamma_t = 1.5\,\text{GeV}$ was used as the reference case. The run-time for the reference sample without *W* decays was 1.7 ms/event, increasing to 3.5 ms/event with *W* decays included. Tests performed on a single 2.7 GHz Intel Core i7 CPU, with clang version 13.0.0 with -O2 optimisation.

# 3 Electroweak Showers

In this section, we discuss the implementation of electroweak radiation in the Vincia parton shower. The realisation in Vincia draws heavily from the formalism set out in [16]. We provide a brief summary of the common points here and discuss the adjustments that have been made to assimilate it with the Vincia QCD shower. A comprehensive description of the QCD shower, including details like its antenna functions, exact phase-space factorisation and kinematic maps may be found in [5, 41, 42].

Vincia's QCD shower is based on the antenna subtraction formalism [43, 44] and allows for the evolution of states with definite helicity [42, 45, 46]. This property is especially important in the electroweak sector due to its chiral nature [15, 16]. However, it does not equate to a complete treatment of spin interference effects. Their inclusion in parton showers has received some attention recently [24, 25]. However, an extension of the algorithms described there to the EW sector would not be straightforward. Spin does not affect emission rates in QCD, but only causes modulation in azimuthal distributions. Practically, this means that one does not need to modify the Sudakov form factor when applying corrections due to spin correlations. This is no longer the case in the EW sector, where the vector boson couplings depend on the spin of the branching particle. The intermediate solution of evolving definite helicities is able to describe such effects, while not requiring an increase in the algorithmic complexity of the shower.

## 3.1 Antenna Functions

To facilitate evolution with polarised particles, branching kernels for every helicity configuration are required. The kernels described in [16] are computed with the spinor-helicity formalism [47], similar to those used in [48, 49] to compute QCD antenna functions. While these branching kernels capture the appropriate quasi-collinear singular structure of electroweak

branchings, they suffer from numerical issues away from the singular limits due to a mismatch with the Vincia shower variables. In the new implementation, they have thus been replaced by antenna functions, bringing the formalism further in line with the QCD sector. While the electroweak antenna functions are defined in terms of the same phase-space invariants as those of the QCD sector, we currently make no attempt to enforce soft coherence in the electroweak sector. In [50], the complete multipole structure of photon emissions was included in a dedicated QED shower without spin-dependence. An extension to the EW sector requires a generalisation of that algorithm to include all EW couplings and helicity dependence, which we reserve for future work.

The calculation of the antenna functions for all final-state EW branchings, as well as vector boson emissions off fermions in the initial state is detailed in A. Crucially, the formalism ensures that its polarisations match up with those used by the HELAS [51] routines implemented in MadGraph5 [52] which Vincia uses to polarise the hard scattering. Special care has to be taken to avoid gauge-dependent relics associated with the Goldstone component of the longitudinal polarisation, which would cause unphysical energy scaling that would violate unitarity if not treated correctly [15–17]. The antenna functions calculated here have been validated to produce the same collinear limits as those of [16].

## 3.2 Evolution Variables and Overestimate Determination

In correspondence with Vincia's QCD shower, pre-branching momenta are denoted by capital letters and post-branching momenta by lowercase letters. The EW shower includes emissions from two final-state momenta, $I, K \to i, j, k$, and from two initial-state momenta, $A, B \to a, j, b$. Other types of emissions, such as those off one initial-state momentum and one final-state momentum, are not included. The current implementation does not attempt to correctly model coherent weak gauge boson emissions, and as such all relevant singularities can be incorporated with the above two types of branchings. The shower formalism is set up in terms of Lorentz-invariant quantities, e.g. $s_{ij} = 2p_i \cdot p_j$. It is convenient to define

$$
\begin{aligned}
m_{ij}^2 &= (p_i + p_j)^2 = m_i^2 + m_j^2 + s_{ij} \\
q_{ai}^2 &= (p_a - p_i)^2 = m_a^2 + m_i^2 - s_{aj}.
\end{aligned}
\tag{8}
$$

The ordering scales are then given by

$$
\begin{aligned}
Q_{\text{FF}}^2 &= p_{\perp,\text{FF}}^2 = \frac{(m_{ij}^2 - m_I^2)(m_{jk}^2 - m_K^2)}{s_{IK}} \\
Q_{\text{II}}^2 &= p_{\perp,\text{II}}^2 = \frac{(m_A^2 - q_{aj}^2)(m_B^2 - q_{bj}^2)}{s_{ab}},
\end{aligned}
\tag{9}
$$

where in practice all initial state momenta are treated as massless. The Sudakov form factor, *i.e.* the probability that no EW branching occurs between scales $Q_n^2 > Q_{n+1}^2$, may then be written as

$$
\Pi_n(Q_n^2, Q_{n+1}^2) = \exp\left(-\sum_{i \in \{n \mapsto n+1\}} \mathcal{A}_i(Q_n^2, Q_{n+1}^2)\right),
\tag{10}
$$

where

$$
\mathcal{A}_i(Q_n^2, Q_{n+1}^2) = 4\pi \int_{Q_{n+1}^2}^{Q_n^2} \alpha(Q^2) a_i(Q^2, \zeta) R_f \, d\Phi_{\text{ant}}.
\tag{11}
$$

Here, $\alpha(Q^2)$ is the running electroweak coupling evaluated at first order and $a_i$ are the antenna functions defined in app. A, written in terms of the ordering scale $Q^2$ and an auxiliary variable $\zeta$. The PDF ratio $R_f$ is given by

$$R_f = \begin{cases} 1 & \text{FF}, \\ \frac{f_a(x_a, Q^2)}{f_A(x_A, Q^2)} \frac{f_b(x_b, Q^2)}{f_B(x_B, Q^2)} & \text{II}. \end{cases} \tag{12}$$

Finally, $d\Phi_{\text{ant}}$ is the parton-shower component of the antenna phase-space factorisation as defined in app. B. The shower distributes its EW branchings according to the probability distribution

$$P_{\text{branch}}(Q_n^2, Q_{n+1}^2) = \frac{d\Pi_n(Q_n^2, Q_{n+1}^2)}{d \log(Q_{n+1}^2)}. \tag{13}$$

Practically, this is sampled using the Sudakov veto algorithm [53–55], where the individual components inside the evolution integral eq. (11) are overestimated by strictly larger and simpler expressions, and branchings get accepted with probability

$$P_{\text{accept}} = \frac{\alpha_i(Q^2)}{\hat{\alpha}} \frac{R_f}{\hat{R}_f} \frac{a_i(Q^2, \zeta)}{a_{\text{trial}}(Q^2, \zeta)}, \tag{14}$$

where $\hat{\alpha}$ is a constant overestimate of the coupling constant, $\hat{R}_f$ is a constant overestimate of the PDF ratio and $a_{i,\text{trial}}(Q^2, \zeta)$ is a simple overestimate of the antenna functions $a_i(Q^2, \zeta)$. The coupling constant $\alpha(Q^2) < \alpha(Q_n^2)$, the latter of which functions as the constant overestimate. Like the full antenna functions, the trial antennae are defined in terms of the Lorentz-invariant dot products. Due to the vast number of possible branchings in the electroweak sector that all come with varying masses, couplings and helicity dependence, the process of determining trial antennae is automated. To that end, parameterised overestimates are defined as

$$a_{i,\text{trial}}^{\text{FF}} = \frac{1}{m_{ij}^2 - m_I^2} \left[ c_1^{\text{FF}} + c_2^{\text{FF}} \frac{1}{x_i} + c_3^{\text{FF}} \frac{1}{x_j} + c_4^{\text{FF}} \frac{m_I^2}{m_{ij}^2 - m_I^2} \right],$$

$$a_{i,\text{trial}}^{\text{II}} = c^{\text{II}} \frac{1}{m_A^2 - q_{ai}^2} \frac{s_{ab}}{s_{AB}} \frac{1}{x_j}, \tag{15}$$

where $x_i$ and $x_j$ are collinear momentum fractions defined in app. A. Note that in the final-final case, a term proportional with the mass $m_I$ appears, but terms proportional with $m_i$ and $m_j$ are absent. The latter terms are not required because the associated mass corrections in the antenna functions tend to be negative, while those that scale with $m_I$ tend to be positive and become dominant when $Q^2 \approx m_I^2$. The trial antenna for initial-initial branchings consists of a single term because it only has to model vector boson emissions. While appropriate values for $c^{\text{II}}$ are easily computed for all flavor and helicity configurations, values of the coefficients $c_1^{\text{FF}}$ through $c_4^{\text{FF}}$ are determined automatically. A large number of branchings is sampled from randomly sampled antenna masses, and the corresponding antenna function is determined for each of them. A suitable trial antenna may then be found by minimising the average distance between the trial- and real antennae while ensuring that for no point in the sample $a_{i,\text{trial}}^{\text{FF}} < a_i^{\text{FF}}$. Such optimisation procedures are instances of *linear programming*, for which many external solvers are available. We make use of the Python package PuLP [56]. The trial evolution integrals eq. (11) are worked out in app. B.

### 3.3 Recoiler Selection

While in Vincia's QCD shower antenna are naturally spanned between colour-connected partons, no such guiding principle exists in the electroweak sector. In fact, since the electroweak

shower makes no attempt to incorporate soft interference effects, the second particle only acts as a recoiler to the collinear emitter. In [16] it was shown that through a probabilistic choice of recoiler selection the effects of recoil of previous branchings may be partially mitigated. In [15], a similar modification was included as a multiplicative factor on the branching kernel. We adopt the strategy of probabilistic choice here, where the probability to select a final-state recoiler $K$ for final-state particle $I$ may be written as

$$P_{\text{rec},I}^K = \frac{\sum_X |A_{X \mapsto IK}(p_I, p_K)|^2}{\sum_{K'=1}^N \sum_{X'} |A_{X' \mapsto IK'}(p_I, p_{K'})|^2} \, , \tag{16}$$

where the functions $A(p_i, p_j)$ are the collinear helicity-dependent $1 \to 2$ branching amplitudes computed in [16]. The sum over $K'$ in the denominator runs over all recoiler candidates, and the sums over $X$ and $X'$ run over all electroweak clusterings of particles $I$ and $K$ or $K'$ respectively. The probability given by eq. (16) gives preference to recoilers $K$ that were most likely to have emitted $I$. Because the final-final kinematic mapping preserves the total antenna momentum, the propagator of the corresponding particle $X$ then remains preserved. Initial state branchings are always set to recoil against the other initial state.

### 3.4 Bosonic Interference

The electroweak sector has the unique feature of introducing interference effects between neutral vector bosons [15, 16, 57]. In [15] a physically accurate but computationally expensive treatment was outlined using evolution of mixed states in density matrices instead of a single definite event. We instead opt for a much simpler treatment, assigning an event weight to the event every time a neutral boson disappears through splitting to fermions or $W$ bosons. For post-branching momenta $p_i$ and $p_j$, the weight associated with the interference between neutral bosons $V_1$ and $V_2$ is given by

$$w_{\text{int}} = \frac{\sum_X |A_{X' \mapsto XV_1}(p_X, p_{ij}) A_{V_1 \mapsto ij}(p_i, p_j) + A_{X' \mapsto XV_2}(p_X, p_{ij}) A_{V_2 \mapsto ij}(p_i, p_j)|^2}{\sum_X |A_{X' \mapsto XV_1}(p_X, p_{ij}) A_{V_1 \mapsto ij}(p_i, p_j)|^2 + |A_{X' \mapsto XV_2}(p_X, p_{ij}) A_{V_2 \mapsto ij}(p_i, p_j)|^2} \, , \tag{17}$$

where $p_{ij} = p_i + p_j$. The pair $V_1$ and $V_2$ corresponds with either a photon and a transversely polarised $Z$ boson, or a Higgs and a longitudinally polarised $Z$ boson. Interference between spin states is not included in correspondence with the rest of the shower formalism. The sum over $X$ runs over all particles that may have emitted the neutral vector bosons through the collinear branchings $X' \to XV_1$ and $X' \to XV_2$.

### 3.5 Resonance Matching

The electroweak shower involves a number of branchings that would normally be associated with resonance decays, such as $t \to bW$, $Z \to q\bar{q}$, etc. In such cases, the EW shower correctly describes the collinear dynamics of these branchings at off-shellness scales that are much larger than the EW scale $Q_{\text{EW}}^2$. At such scales, the resonance masses are subleading corrections to the regular collinear factorisation dynamics. However, at scales close to the resonance width, resonance decays are more correctly described by a Breit-Wigner distribution. Pythia's default method of treating resonance decays is to sample the resonance mass from a Breit-Wigner distribution upon production. Vincia follows this procedure, making use of helicity- and mass-dependent decay widths that contain $\mathcal{O}(\alpha_s)$ corrections. These widths, as well as the sampling procedure, is detailed in app. C.

Resonances may be produced as part of the hard scattering, or by the EW shower. In either case, the EW shower may sample resonance-like branchings at scales sufficiently above the EW scale. The Breit-Wigner-sampled masses on average differ from the on-shell mass only by $\mathcal{O}(\Gamma)$,

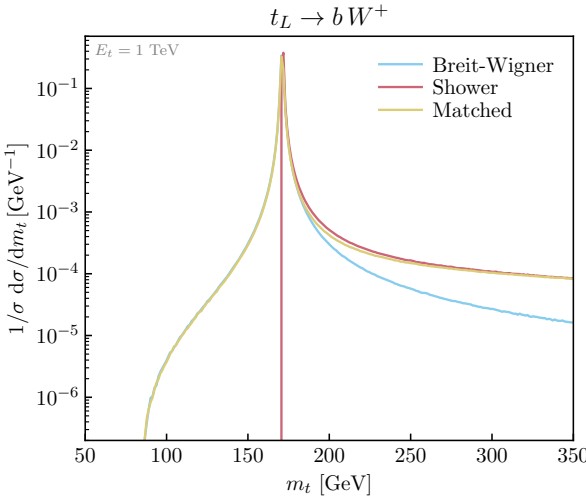

Figure 4: The mass spectrum of a 1 TeV left-handed top quark decay as generated by the Breit-Wigner distribution only (blue), the EW shower (red) and the numerically matched procedure (yellow). The matched spectrum shows a smooth transition between the Breit-Wigner region close to the pole mass, and the parton-shower distribution far away from it.

and thus do not affect the EW shower significantly. The resonance decay-like branchings are suppressed by a factor $Q^4/(Q^2 + Q_{EW}^2)^2$, such that the shower branching probability smoothly vanishes at scales close to $Q_{EW}^2$. Then, when the shower evolution reaches the scale associated with the offshellness of the resonance, given by eq. (4), without branching it, the resonance is decayed according to the procedure outlined in app. C. This ensures a smooth matching between the two descriptions, as is illustrated in fig. 4 for the case of a left-handed top decay. The value of $Q_{EW}^2$ is implemented as a tunable parameter in this context. All high-energy SM resonances produced by the EW shower show the behaviour illustrated in fig. 4, where the EW shower dominates the Breit-Wigner shape at high invariant mass. This means that the exact treatment of running width effects far away from the peak, which are prone to gauge violations, are of little practical importance. The interleaving procedure described in the previous section is available for all resonance decays, whether they are produced or decayed by the EW shower or otherwise.

## 3.6 Overlap Veto

When the parton shower generates both QCD and EW branchings, care has to be taken to avoid double-counting the approximated matrix element of certain topologies. This double-counting may occur when different Born-level events can populate the same phase-space point after shower branchings. For example, a final state $VVj$ may be reached by starting from either a $VV$ born-level state and adding a QCD initial-state branching, or by starting from a $Vj$ born-level state, and adding an EW branching. A solution to this issue was described in [13] and is included in the Pythia EW shower in the case of a single vector boson. Here, we generalise this procedure to be universally applicable.

Double-counting can be avoided by ensuring that the QCD and the EW showers only populate the regions of phase space that they are most accurate in. For instance, in the previously mentioned case of $VVj$ where the two vector bosons are collinear, the path of a $Vj$ + an EW emission should be preferred as the EW shower models collinear vector boson emissions accurately. On the other hand, if the jet is soft, the preferred path is $VV$ + a QCD emission off the

initial state. The above procedure can be implemented generally by introducing a resolution measure that indicates which shower most accurately describes a particular phase space point. Note that this choice of resolution measure is not unique, and any choice that correctly separates the singular regions describes by both showers will suffice. Different choices will lead to different behaviour in intermediate regions where neither shower provides an accurate description.

Here, we choose to follow [13] and make use of a resolution measure inspired by the one used by the $k_t$ jet algorithm [58], generalised to account for particle masses, defined as

$$
d_{ij} = \begin{cases} \min\left(p_{t,i}^2, p_{t,j}^2\right) \frac{\Delta_{ij}}{R} & \text{Final-state}, \\ p_{t,i}^2 & \text{Initial-state}, \end{cases} \tag{18}
$$

where $p_t^2 = k_t^2 + |m_i^2 + m_j^2 - m_I^2|$, $k_t$ is the transverse momentum, $\Delta_{ij}$ is the angular distance between $i$ and $j$. The parameter $R$ determines the relative weight associated with initial-state and final-state emissions, again affecting behaviour in the non-singular intermediate regions. It is left as a tunable parameter with default value equal to the implementation of [13]. The mass corrections are included to more accurately reflect Vincia's ordering scale. For instance, in a branching like $W^{\pm} \to W^{\pm}Z$, the appropriate measure is $p_t^2 = k_t^2 + m_z^2$. The absolute value is added to prevent the measure from dropping below zero. It is only relevant in resonance-like branchings where $m_I > m_i, m_j$, in which case no overlap between the QCD and EW sector exists anyway.

The resolution measure $d_{ij}$ is small in all singular regions of phase space for both QCD and EW emissions, and can thus be used to define a veto procedure. More specifically, if, for instance, the shower generates a QCD emission, eq. (18) is evaluated for that branching, as well as for all possible $2 \to 1$ EW clusterings in the post-branching event. Then, the QCD branching is accepted if $d_{ij}^{\text{QCD}} < \min\left(d_{ij}^{\text{EW}}\right)$, and vetoed otherwise. The reverse procedure is also applied if an EW branching is generated. This procedure effectively sectorises the QCD and EW showers, ensuring that every phase-space point is only populated by either a QCD or EW shower history. Note that such a sectorization is also relevant for the construction of shower histories in the context of matrix element merging. Vincia's QCD shower is itself sectorized [42], ensuring only a single QCD shower history is associated with every phase space point. In a future version, the EW shower may also be sectorized, leading to a single shower history through both showers in combination with the overlap veto.

# 4 Validations, Results and Discussion

In this section, we present several validations and results concerning both the implementation of interleaved resonance decays, as well as the EW shower.

## 4.1 Interleaved Resonance Decays in $ee \to t\bar{t}$

We take the Vincia sector-antenna shower [42] implemented in Pythia 8.306 [4] as our baseline for illustrating the effects of interleaving. (For the non-interleaved case, a detailed study of Vincia's treatment of radiation in top-quark decay can be found in [38].)

In all of the figures presented in this section, we take the conventional (sequential) treatment of resonance decays as our baseline (blue), and compare with the new interleaved method with a fixed scale equal to the width $\Gamma_0$ (red), or a dynamic scale choice, $P(m)$, given by the inverse-propagator distribution, eq. (4) (yellow). We note that the latter treatment is now the default in Vincia since Pythia version 8.304 (while Pythia's simple showers retain the sequential treatment as default).

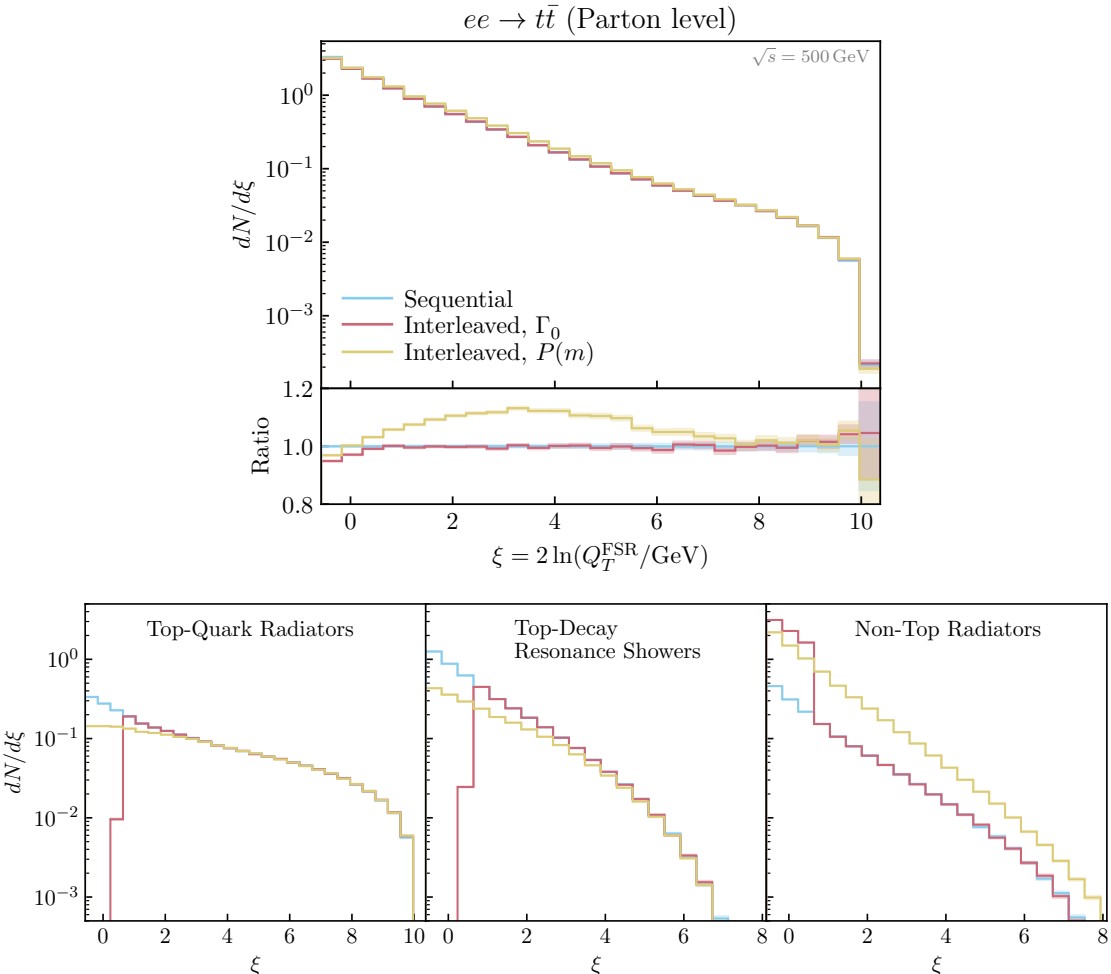

Figure 5: Vincia FSR branchings as functions of $\xi = \ln p_\perp^2$. *Upper Pane:* summed over all radiators. *Bottom Panes:* separated into the three radiation classes described in the text. The colour coding in the bottom panes is the same as in the top pane. Here, and in future figures, shaded regions indicate statistical uncertainties.

As a first simple test case, we consider $e^+e^- \to t\bar{t}$ at $\sqrt{s} = 500\,\text{GeV}$, for the same mass values as were used in [35], $m_t = 173.3\,\text{GeV}$ and $m_W = 80.385\,\text{GeV}$. To focus on the radiation emitted from the top quarks (and their decays), we keep the $W$ bosons stable, and QED bremsstrahlung from the incoming beams is switched off.

In fig. 5, we show the spectra of FSR branchings as a function of $\xi = \ln p_\perp^2$, where $p_\perp$ is Vincia's shower evolution scale. For reference, $\xi(m_t) = 10.3$, $\xi(20\,\text{GeV}) = 6$, $\xi(2\Gamma_0) = 2.2$, and $\xi(\Gamma_0) = 0.8$, with the top-quark width $\Gamma_0 = 1.5\,\text{GeV}$. The upper pane includes all branchings, regardless of where in the shower they occur, while the three lower panes separate the contributions from (left) emissions off top quarks, (middle) branchings during the top-decay resonance showers, and (right) all other branchings. Starting with the lower left-hand pane of fig. 5, the rate of emissions coming directly from $t$ and/or $\bar{t}$ quarks is unchanged for large $\xi$ (where this is the dominant component of the spectrum). At low $\xi$, the fixed scale choice produces a sharp cutoff at $\Gamma_0$, while the dynamic choice produces a tapering off of the emission rate as the top quarks gradually disappear from the evolution. In the middle pane, we see similar effects inside the top-quark resonance showers. In the sequential case, these continue all the way to the QCD shower cutoff, while they terminate at a higher (fixed or dynamic) scale in

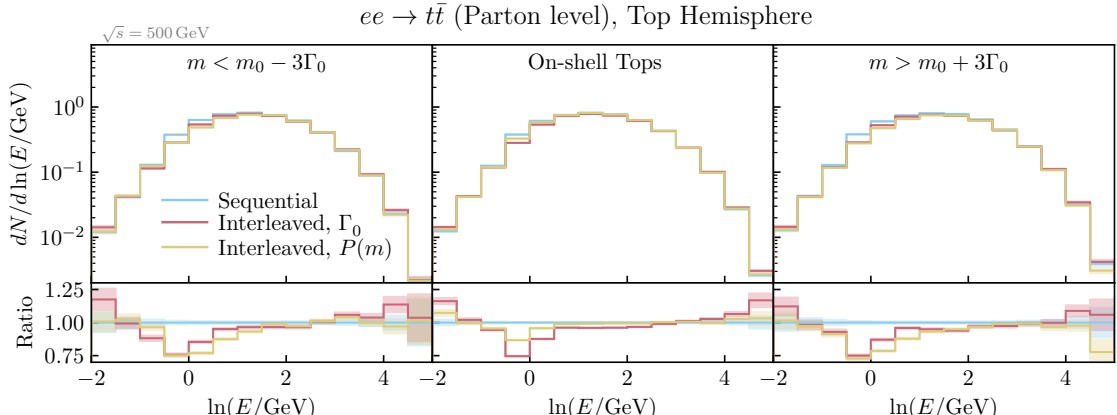

Figure 6: Logarithmic energy spectrum of partons emitted in the top hemisphere, at the shower cutoff scale, for (left) low-mass tops with $m < m_0 - 3\Gamma_0$, (middle) on-shell tops with $|m - m_0| < \Gamma_0/2$, and (right) high-mass tops with $m > m_0 + 3\Gamma_0$. Note that the spectra are all normalised to the cross sections of their respective regions.

the interleaved cases. In other words, the first two lower panes of fig. 5 illustrate that, in the interleaved treatment the top quarks themselves (and the component of their decay systems that is still treated as factorised from the rest of the event) are only relevant at evolution scales greater than their offshellnesses.

The rightmost lower pane of fig. 5 shows that this difference is almost perfectly made up for by an increase in radiation not associated directly with the top quarks (nor with their resonance showers). In the interleaved case, this is the component that dominates the evolution at low $\xi$, while in the sequential case those branchings still take place within the top resonance showers. Interestingly, for the dynamic scale choice, the increase in this component persists up to quite high scales. For reference, $\xi = 5.4$ corresponds to a scale of order ten times the width. This is due to the constraint of preserving the invariant mass of the top-decay system being lifted quite early on in the evolution for highly off-shell tops and the opening up of a larger phase space, especially relevant on the low-mass tail.

Although the relative contributions of each of the three classes of branchings shown in the left-hand pane thus change by substantial amounts, it is reassuring to note that, at the summed level, the total emission rate illustrated in the main (upper) pane of fig. 5, only changes by small amounts. We note that the total emission rate of very soft emissions, for scales below $\sim 1\,\text{GeV}$ ($\xi \sim 0$) at the extreme left-hand edge of the plot, drops a little. We regard this as a reasonable physical consequence of preventing the top-production and top-decay systems from both radiating at scales below the width; now only the decay products can do that. With the dynamic scale choice, there is moreover also a $\sim 10\%$ enhancement of radiation at scales up to an order of magnitude larger than the width, due to the aforementioned opening-up of the phase space after exiting the resonance shower. We believe this is an interesting effect which exemplifies a non-trivial interplay between the two resummations.

The evolution-scale spectrum itself is of course model dependent and not physically observable. We show it mainly to illustrate the scale progression in the underlying algorithm. As an intermediate step before showing physical hadron-level observables, fig. 6 shows the (logarithm of the) energy spectrum of partons emitted into the top-quark hemisphere, at the shower cutoff scale (by default 0.75 GeV in Vincia), excluding $b$ quarks and $W$ bosons so that the spectrum only reflects the emitted partons. The left- and right-most panes show the spectra for off-shell tops with $m - m_0 < -3\Gamma_0$ and $m - m_0 > 3\Gamma_0$ respectively (each corresponding to about 5% of the total number of top quarks), while the central pane shows it for top quarks

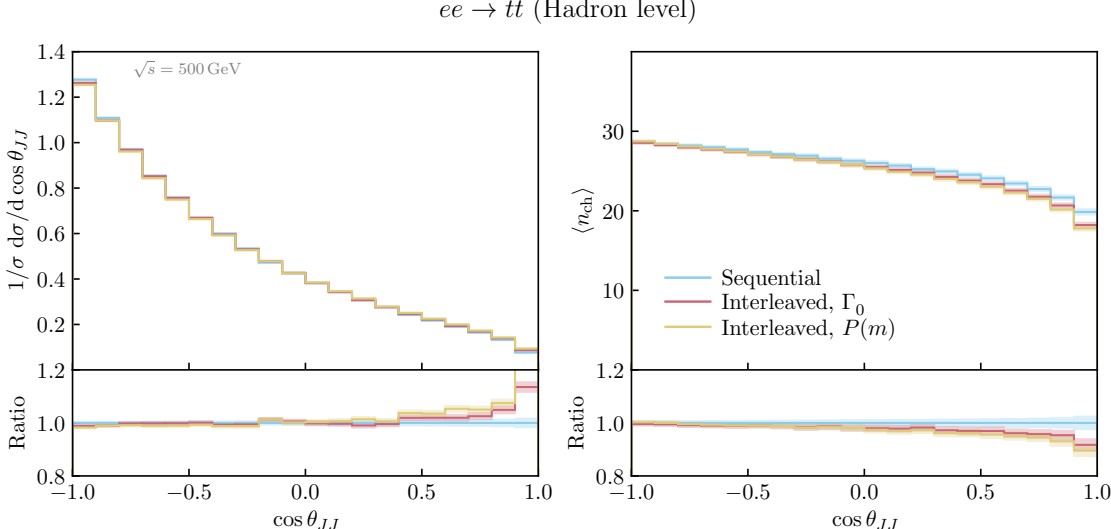

Figure 7: Distributions of the angle between the two clustered jets (left), and the average charged multiplicity as a function of that angle (right), in leptonic $e^+e^- \to t\bar{t}$ events at $\sqrt{s} = 500$ GeV.

with $|m - m_0| < \Gamma_0/2$ (corresponding to about half of all top quarks). The emission rates are all normalised to the total cross section in their respective regions. The dominant feature is that interleaving suppresses radiation at energies at and just below the width. For the dynamic choice of interleaving scale, this is more pronounced for off-shell tops (left and right panes) than for on-shell ones (middle pane), while for the fixed scale choice it is independent of the offshellness, as expected. Interestingly, the fixed scale choice in particular also results in a small *increase* in radiation at very low scales ($\xi \sim -2$ corresponding to $E \sim m_\pi$) though it is doubtful whether this would have any observable consequences.

Turning to hadron-level quantities, we first consider two observables proposed in [9], defined by clustering all particles in the event (omitting any that originate from the $W$ decays) into exactly two jets (using the $e^+e^-$ $k_t$ algorithm [59] via FastJet [60]), which are stand-ins for the Born-level $b$ quarks. This is most appropriate near the $t\bar{t}$ threshold where there is no radiation from the $t\bar{t}$ pair before decay. In that case, neglecting spin correlations between the two decays, the two jets should be distributed independently and isotropically, *i.e.* with a uniform distribution in $\cos\theta_{JJ}$. In our case, we are considering a situation somewhat above threshold, with a CM energy of 500 GeV corresponding to a Lorentz boost factor for each of the top quarks of $\gamma = 1.45$. The $\cos\theta_{JJ}$ distributions shown in the left-hand pane of fig. 7 are therefore peaked towards -1, but there is still an interesting shift towards less strong anti-collimation when interleaving is enabled, which is especially pronounced in the (suppressed) region where the two jets end up with a relative angle of less than 60 degrees ($\cos\theta_{JJ} > 0.5$). This agrees qualitatively with the conclusions of [9].

In the right-hand pane of fig. 7, we show how the average charged multiplicity depends on $\cos\theta_{JJ}$. Not surprisingly, the largest multiplicities occur when the two jets are approximately back to back, while fewer particles are produced when the two jets are closer to each other. This trend becomes stronger when interleaving is enabled, again in line with what was found in [9].

We round off our discussion of $e^+e^- \to t\bar{t}$ by considering the hadron-level energy distribution as a function of the angle from the $b$ quark (which we take as a proxy for the $b$-jet). This is somewhat motivated by the studies in [10], which however used the top direction as the

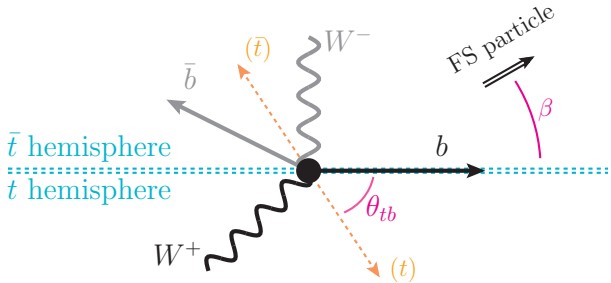

Figure 8: Illustration of the definitions of the angles $\theta_{tb}$ and $\beta$, and the definitions of the $t$ and $\bar{t}$ hemispheres.

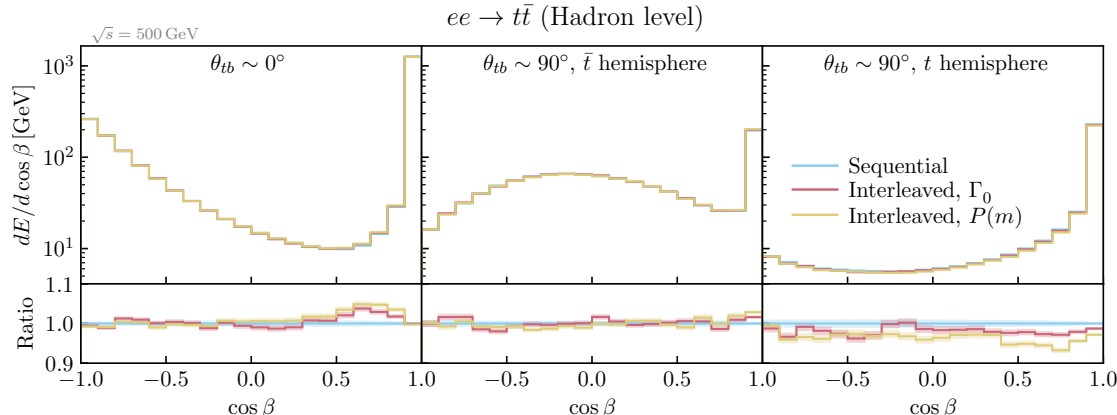

Figure 9: Hadron-level energy distribution in leptonic $e^+e^- \to t\bar{t}$ events at $\sqrt{s} = 500\,\text{GeV}$, as a function of the angle from the $b$ jet, for (upper pane) events in which the $b$ quark from the top-quark decay was emitted "forwards" (with $\cos\theta_{tb} > 0.9$) and (lower panes) at right angles (with $\sin\theta_{tb} > 0.9$), with the latter divided into two hemispheres, *cf.* fig. 8.

reference. We use the $b$-quark direction instead since we are still relatively close to threshold and this places the peak of the main observed jet at a well-defined location in the plots. To focus on the top-associated particle production, we consider fully leptonic events, and remove the leptons from the $W$ decays from the energy distributions. The coordinate system is illustrated in fig. 8. We first consider the case when the angle between the $t$ and $b$ directions, $\theta_{tb}$, is small, $\cos\theta_{tb} > 0.9$. Then, the $t$ and $b$ directions are approximately degenerate, and the $\bar{t}$ is at 180°. The energy distribution for these events, as a function of the angle to the $b$ quark, is shown in the left-hand pane of fig. 9. In the interleaved cases, there is a slight ($\sim 5\%$) increase in the energy deposited in the outer regions of the $b$ jet, relative to the sequential decay.

In the middle and right-hand panes of fig. 9, we consider events with a large angle between the $t$ and $b$ directions, $\sin\theta_{tb} > 0.9$. We divide these events into two hemispheres, as illustrated in fig. 8. In the middle pane, we show the angular distribution in the $\bar{t}$ hemisphere, which contains the flight direction of the $\bar{t}$. This hemisphere thus tends to contain the $\bar{b}$ jet, with is smeared over a large angular region due to varying event kinematics and the fact that the $\bar{t}$ is not highly boosted at $\sqrt{s} = 500\,\text{GeV}$. No significant effects of interleaving are evident in this hemisphere. The $t$ hemisphere, on the other hand, is relatively free from contamination of the $\bar{b}$ jet, and here we see a small suppression of the energy density at basically all angles.

Thus, we find that there can be kinematics-dependent modifications of up to $\sim 5\%$ to the energy flow in the event, here illustrated taking the $b$-quark direction as reference axis.

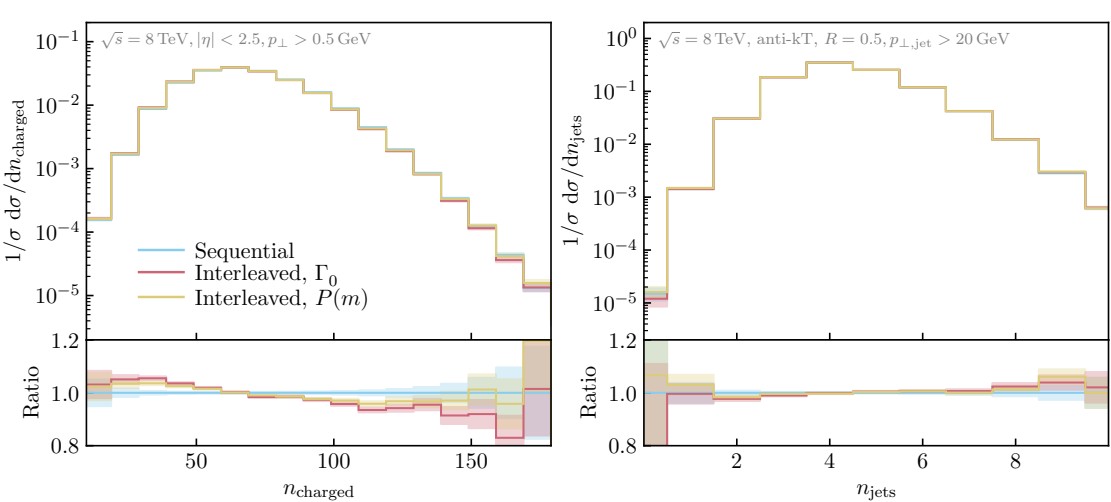

Figure 10: Multiplicity of charged particles (left) and jets (right) in semi-leptonic $pp \to t\bar{t}$ events at $\sqrt{s} = 8000\,\text{GeV}$.

## 4.2 Interleaved Resonance Decays in $pp \to t\bar{t}$

As our final example of the effects of interleaved resonance decays, we take $pp \to t\bar{t}$ at $\sqrt{s} = 8\,\text{TeV}$, repeating the analysis of semileptonic events that was used to estimate colour-reconnection effects on the hadronically reconstructed top-quark mass in [35].

In the left-hand pane of fig. 10, we show the multiplicity of central charged particles, with $|\eta| < 2.5$ and $p_\perp > 0.5\,\text{GeV}$. We see that both of the interleaved options produce a slight reduction in the multiplicity of charged tracks relative to the sequential decay treatment. The multiplicity of anti-$k_t$ jets [61] with $R = 0.5$ and $p_\perp > 20\,\text{GeV}$, shown in the right-hand pane, is not affected.

The result of the primitive hadronic $W$ and $t$ mass reconstruction of [35] is shown in fig. 11. The anti-$k_t$ jets found above are combined to form jet pairs which are accepted as $W$ candidates if they have a combined invariant mass in the vicinity of the $W$ mass ($|m_{jj} - m_W^{\text{pole}}| < 5\,\text{GeV}$). The best such $W$ candidate is then combined with a third jet and accepted as a top candidate if $|m_{jjj} - m_t^{\text{pole}}| < 20\,\text{GeV}$. The difference between the invariant mass of the best $W$ candidates in each event and the $W$ pole mass value is shown in the left-hand pane of fig. 11, with interleaving producing no significant differences in the distribution. The slight increase at higher masses translates to an increase in the hadronically reconstructed $W$ mass of 35 MeV (with a statistical MC uncertainty of 6 MeV) for both of the interleaving options.

The right-hand pane of fig. 11 shows difference between the hadronically reconstructed top-quark mass and the pole-mass value. Again, reassuringly, there is essentially no difference visible at the resolution scale of the plot. Nevertheless, the directly calculated mean of the $\Delta m_t$ distribution decreases by about 140 MeV (with a statistical MC uncertainty of 25 MeV) when interleaving is switched on (contrary to the $W$ mass which changed in the opposite direction). This change is somewhat larger than the 30-MeV figure obtained for $e^+e^- \to t\bar{t}$ in [9], and may have relevance for high-precision top-quark mass measurements at LHC. A more elaborate assessments with full-fledged top-mass extraction methods, including the effects of in-situ calibration, is beyond the scope of this work but would be well-motivated.

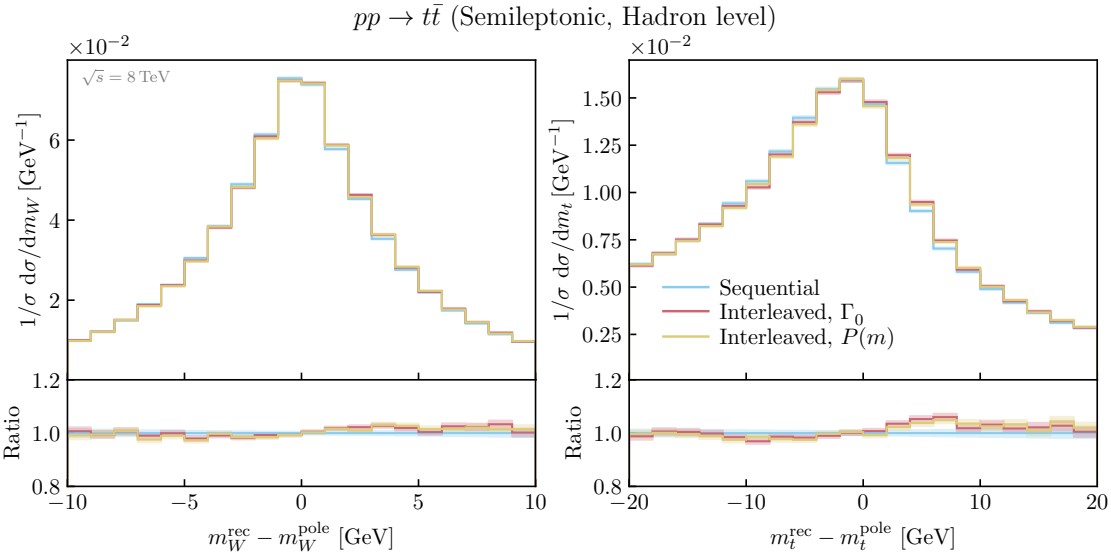

Figure 11: Difference between the reconstructed and the pole masses of hadronically decaying $W$ bosons (left) and top quarks (right), in semi-leptonic $pp \to t\bar{t}$ events at $\sqrt{s} = 8000$ GeV.

## 4.3 Validation of the EW Overlap Veto

In this section, we validate the implementation of the overlap veto procedure, as well as the correct implementation of triple-vector boson interactions in the EW shower. To that end, we compare the direct matrix element for $pp \to VVj$ at 14 TeV with the shower prediction. Here, the vector bosons include $Z$, $W^+$ and $W^-$ and the jet is a gluon or a light quark. Fig. 12 shows the comparison of the leading-order matrix element with several shower histories, as a function of the angular separation between the vector bosons $\Delta R_{VV}$. This observable serves as a natural separator between the regions of phase space where the individual shower contributions should approximate the matrix element well. For small angular separations, the vector bosons are collinear and the EW shower should perform best, while at large angular separation the vector bosons are back-to-back, and the QCD shower should be preferred. To allow for sufficient phase space for the EW shower to radiate in, the hard scattering is ensured to be highly energetic by requiring $0.5$ TeV $< p_{\perp,\text{jet}} < 1$ TeV. The dashed lines show the individual contributions of $pp \to VV$ with a QCD emission and $pp \to Vj$ with an EW emission, and the solid blue line shows their sum. When the overlap veto is not enabled, the sum of contributions clearly overshoots the exact matrix element significantly. On the other hand, when the overlap veto is enabled, the summed Vincia shower prediction matches the matrix element very well and the showers correctly cover their associated regions of phase space.

## 4.4 Fragmentation in Heavy Dark-Matter Decays

As another test of the EW shower implementation, we consider the computation of the prompt decay spectra of heavy dark matter that decays to Standard Model particles. If such dark matter particles are indeed heavy enough, an EW shower develops as it decays, and the shower products may appear on earth in the form of cosmic rays. Recently, these decay spectra were computed by Bauer, Rodd and Webber in [62], expanding upon previous work [63, 64]. Their methods involve a numerical evolution of DGLAP equations above the EW scale in the unbroken phase of the Standard Model. The result is then matched to the broken phase at the EW scale, after which the rest of the evolution is performed with Pythia, until only stable particles

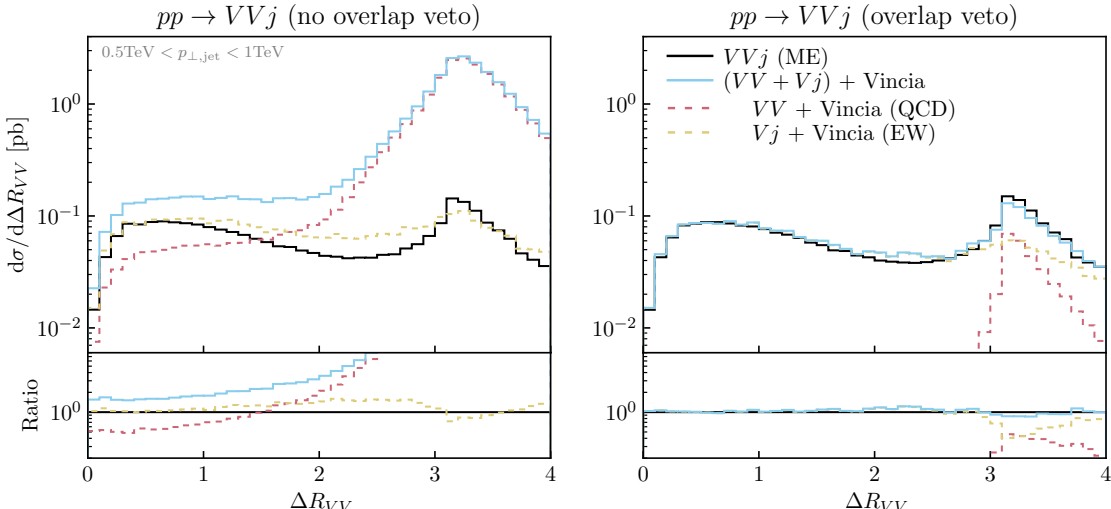

Figure 12: A comparison between the exact leading order matrix element for $pp \to VVj$ (solid black) and different shower approximations at 14 TeV. These are $pp \to VV$ + Vincia QCD (dashed red) and $pp \to Vj$ + Vincia EW (dashed yellow), as well as their sum (blue). On the left-hand side the overlap veto is disabled, leading to an overcounting of the exact matrix element, while it is enabled on the right-hand side, where the matrix element is reproduced accurately.

$S = \{\gamma, e^\pm, p^\pm, \nu_{e,\mu,\tau}, \bar{\nu}_{e,\mu,\tau}\}$ remain. With the inclusion of all possible collinear branchings in Vincia's EW shower, the same physics content should be included. However, the methods of [62] allow for evolution up to dark matter masses up to the Planck scale, while Pythia and Vincia can go up to $\mathcal{O}(100 \text{ TeV})$ before issues related to numerical precision start to appear. Fig. 13 shows a comparison between their results and the Vincia and Pythia predictions for the energy spectrum of a number of stable particles in the decay of a 200 TeV dark matter particle to electron-neutrinos. Already at this energy, the difference between the Pythia prediction and the two other results is apparent. It is again caused by the missing physics content of Pythia's EW shower, which includes the absence of triple-vector boson interactions and a treatment of spin. The difference is particularly striking in the hard photon spectrum, where the Pythia prediction drops off rapidly while the other two lines show a characteristic bump. Another striking feature is the fact that Vincia shows consistently relatively decreasing soft spectra, while Pythia appears to match up with the results of [62] up to a vertical shift. This is likely caused by the fact that [62] use Pythia to do the low-scale evolution, while in the former case the complete evolution is performed by Vincia. Furthermore, there are significant differences between the treatment of [62] and Vincia's EW shower. These include treatment of spin interference, a different treatment of soft interference and the matching procedure at the EW scale, which is not required in Vincia as it performs all evolution in the broken description of the Standard Model.

## 4.5 Electroweak Sudakov Logarithms

The previous two sections have illustrated the ability of the EW shower to correctly incorporate real EW corrections. In this section, we instead focus on the associated virtual corrections. Virtual EW Sudakov logarithms are known to have large effect in the hard tails of observables already at the LHC, and in particular at future colliders. While such corrections are regulated by the EW scale, they are physically relevant without the inclusion of the associated real corrections, because those lead to different experimental signatures. Much work has already

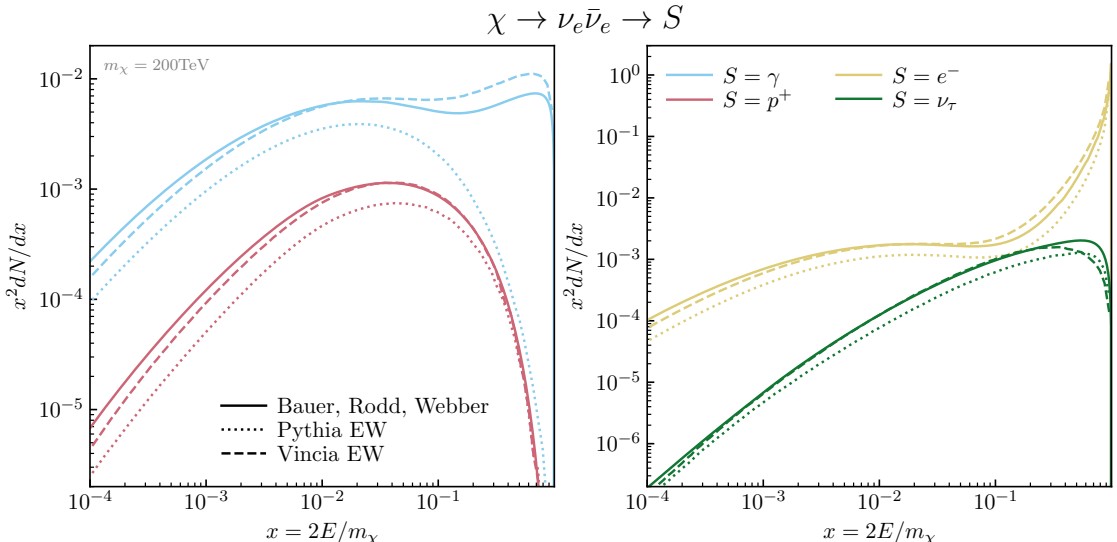

Figure 13: Comparison of the energy spectra of the stable particles $\gamma$ (blue), $p^+$ (red), $e^-$ (yellow) and $\nu_\tau$ (green) in the decay $\chi \to \nu_e \bar{\nu}_e$ with $m_\chi = 200$ TeV, as predicted by Bauer, Rodd and Webber [62] (solid), Pythia's EW shower (dotted) and Vincia's EW shower (dashed).

been done on the analytic calculation and resummation of these corrections [65–73]. Here, we illustrate that they may also be calculated with Vincia's EW shower.

Fig. 14 illustrates the size of the negative virtual EW corrections in the process $pp \to ZZ \to e^+e^-\mu^+\mu^-$ at 14 TeV and 100 TeV as a function of $p_{\perp,Z}$. The EW shower incorporates these virtual corrections through its unitary nature. When it is enabled, some $Z$ bosons will, for instance, branch to $W^+W^-$, or another $Z$ may be radiated from the initial state. As a result, events without such corrections effectively get weighted by the EW no-branching probability where the virtual corrections are exponentiated. Fig. 14 shows these virtual corrections at the Monte Carlo level, as well as at the fiducial level, where the cuts

$$65 \text{ GeV} < m_Z < 115 \text{ GeV}, p_{\perp,l} > 25 \text{ GeV and } |\eta_l| < 3.5 \qquad (19)$$

are applied, to emphasise that they arise as a result of the different experimental signatures of the real corrections. At 14 TeV, the corrections reach $-30\%$ towards the highest end of the spectrum, while they go down to $-70\%$ at 100 TeV.

Fig. 15 compares the run time penalty as a function of $p_{\perp,Z}$ for activating the EW shower in $pp \to ZZ \to e^+e^-\mu^+\mu^-$ at 100 TeV. As might be expected, this penalty increases with $p_{\perp,Z}$ up to a factor of two as a larger phase space opens up for EW radiation in the final state, the products of which might in turn induce further activity.

## 5 Summary and Conclusions

We have presented two interrelated extensions of the Vincia shower framework in Pythia 8, introducing interleaved resonance decays and electroweak shower branchings, respectively, in the perturbative evolution. The latter is based on the formalism presented by one of us in [16], while the former is a new proposal that shares some features with earlier work by Khoze and Sjöstrand [9].

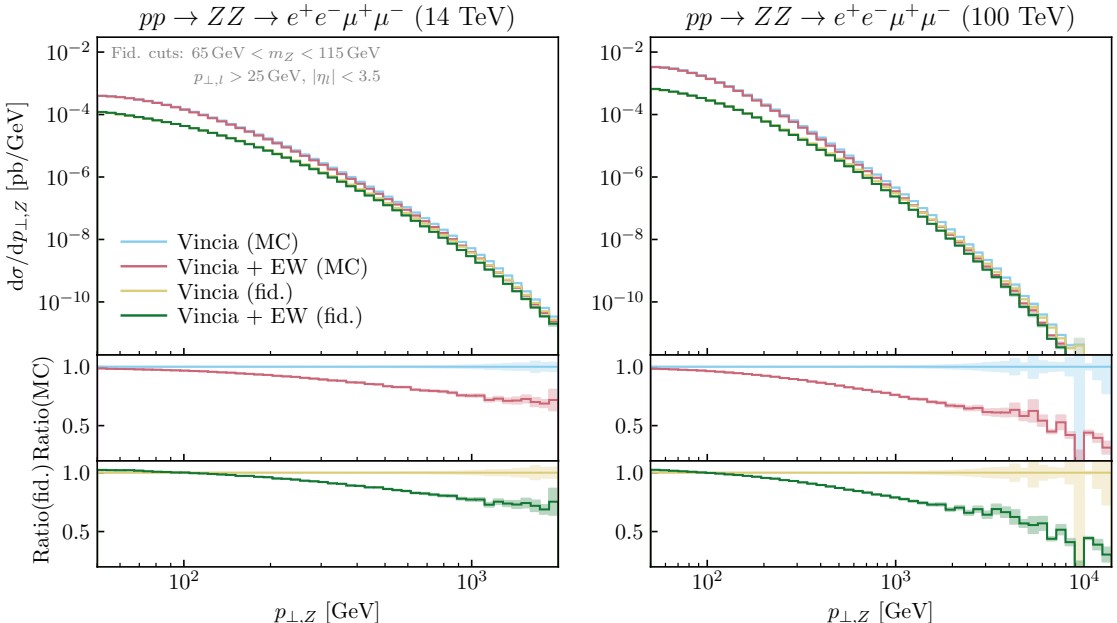

Figure 14: An illustration of the size of the EW virtual Sudakov logarithms generated by Vincia's EW shower in the process $pp \rightarrow ZZ$ at 14 TeV (left) and 100 TeV (right). Shown are the leading-order result without EW shower (blue) and with the EW shower (red) at the Monte Carlo level, as well as those same predictions at the fiducial level (yellow and green respectively).

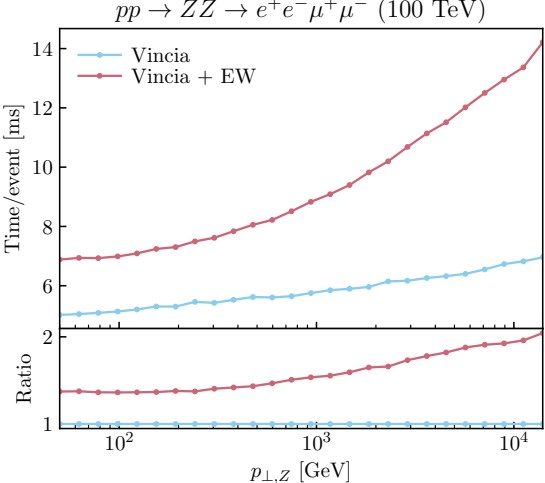

Figure 15: Relative time consumption to generate $pp \rightarrow ZZ \rightarrow e^+e^-\mu^+\mu^-$ at 100 TeV as a function of $p_{\perp,Z}$, without EW shower (blue) and with EW shower (red). Tests were performed on a single M1 Pro core with clang version 13.0.0 with -O2 optimisation.

When decays are interleaved with the final-state shower evolution, each unstable resonance is assigned an offshellness scale, which, in our implementation, can either be chosen to be static (e.g., the width, $\Gamma_0$) or dynamic (e.g., a Breit-Wigner distributed offshellness). When the overall evolution reaches this scale, the resonance is removed from the event and replaced by its decay products; these are then subjected to a "resonance shower" which conserves the

total 4-momentum of the decay system and thereby preserves the shape of the original (Breit-Wigner) resonance-mass distribution. This shower starts at the mass of the decaying resonance and ends when the evolution again reaches the offshellness scale, after which the resonance-decay plus shower products are merged with the production process and the evolution is continued starting from the offshellness scale. In this last step, any resonance-final antennae [38] are replaced by corresponding (non-resonant) final-final ones, which allows for the possibility for further radiation to generate out-of-resonance recoils which can distort the Breit-Wigner shape. The procedure also results in a suppression of low-frequency radiation from short-lived particles. We regard these features as making good physical sense and in line with conclusions of previous studies [9–11,27,29]. Further properties of our model include that cascade decays can be nested recursively when successive offshellness scales are increasing, and the virtue of enabling a robust and relatively simple matching between EW shower branchings and EW resonance decays.

As a complement to the more sophisticated treatment of interleaved resonance decays, the implementation of a full-fledged EW shower is now part of Vincia. It relies heavily on the helicity shower formalism set out in [45,46] to model the chiral nature of the EW sector. The EW shower includes all possible EW collinear splittings, including those originating from triple-vector boson interactions or coupling to the Higgs. These were notably previously missing from Pythia's default EW shower [13]. A number of interesting features appear in the EW sector, including neutral boson interference effects and the matching of resonance-like branching to a Breit-Wigner distribution. Our implementation ensures that such branchings that occur at large offshellness are modelled by the EW shower, while mirroring Pythia's Breit-Wigner-based treatment at small offshellness. Note that the interleaving of resonance decays becomes especially relevant at the large offshellness values probed by the EW shower. Finally, when both the QCD and EW showers are enabled, a risk of double-counting appears due to the possibility of reaching identical states from different Born configurations. To avoid this, a generally-applicable veto procedure was introduced that avoids any overlap and ensures every phase space point is populated by the most accurate shower.

The effects of interleaved resonance decays are particularly relevant for coloured resonances, such as top quarks, but also apply in the QED sector, to electrically charged resonances such as $W$ bosons. In the latter context we note that Vincia has a fully coherent treatment of QED radiation, including multipole interference effects [50,74]. It is, however, not yet possible to enable the new EW shower at the same time as the fully coherent QED module, as it does not support treatment of helicity-dependent photon emissions. One currently has to choose which functionality is most important for the study at hand, with the coherent QED module being the one that is enabled by default.

Case studies in the context of $t\bar{t}$ production at $e^+e^-$ and $pp$ colliders were presented in sec. 4. For the most part, these validated the expectation of relatively modest effects, due to the comparative smallness of the SM top-quark width. Nevertheless, we found a shift of of about 140 MeV on a (primitive) reconstruction of the invariant mass of hadronically decaying tops in $pp$ collisions, which may be worth investigating further, with more full-fledged analysis techniques.

Sec. 4 also includes a validation of the overlap veto of the EW shower, where the $VVj$ matrix element is compared to the shower approximations $VV +$ QCD and $Vj +$ EW. The results show that the EW shower correctly includes the triple-vector boson interactions, and that the overlap veto is necessary to avoid overshooting the three-body matrix element. Furthermore, a comparison to the work of [62] was shown, where the energy spectra of the decay high-mass dark matter to SM particles was computed analytically. We found that, while there are still major differences in the modelling of the physics, Vincia's EW shower closely matches their results. Finally, the size of virtual EW Sudakov logarithms in $pp \to ZZ$ were investigated

at the LHC, as well as at future collider energies. The results show that, as has long been known [65–73], these effects are especially large at future collider energies, and an EW shower includes them systematically.

Potential avenues for future research in the area of interleaved resonance decays include studying the formal interplay between the Breit-Wigner and parton-shower resummations and the modelling non-perturbative effects in the tail of long-lived coloured resonances. On the EW shower side, possible ways forward are the enabling of coherent QED radiation in conjunction with the EW shower and especially assessing its logarithmic accuracy with the modern techniques explored in [75–77] and comparing to the work of e.g. [78, 79].

# Acknowledgements

We would like to thank Philip Ilten for his hard work in helping to integrate the EW shower into the Pythia codebase. R.V. acknowledges support by the Dutch Foundation for Fundamental Research of Matter (FOM) via program 156 'Higgs as Probe and Portal' and by the UK Science and Technology Facilities Council (STFC) via grant award ST/P000274/1. H. B. is funded by the Australian Research Council via Discovery Project DP170100708 — "Emergent Phenomena in Quantum Chromodynamics". This work was also supported in part by the European Union's Horizon 2020 research and innovation programme under the Marie Sklodowska-Curie grant agreement No 722105 — MCnetITN3.

# A   Electroweak Antenna Functions

This appendix provides some details regarding the calculation of the EW helicity-dependent antenna functions, and lists all nonvanishing configurations. Helicities are denoted by $\lambda = \pm 1$ for fermions and transversely polarised vector bosons, and by 0 for scalars and longitudinally polarised vector bosons. The antenna functions are computed by using the spinor-helicity formalism [47] to compute the corresponding $1 \rightarrow 2$ branching processes. Massive Dirac spinors of helicity $\lambda = \pm 1$ are defined as

$$u_\lambda(p) = \frac{1}{\sqrt{2p\cdot k}}(\slashed{p} + m)u_{-\lambda}(r) \text{ and } v_\lambda(p) = \frac{1}{\sqrt{2p\cdot k}}(\slashed{p} - m)u_\lambda(r), \tag{20}$$

while polarisation vectors for massive gauge bosons of helicities $\lambda = \pm 1$ and 0 may be written as

$$\epsilon_\lambda^\mu(p) = \pm \frac{1}{\sqrt{2}}\frac{1}{2p\cdot r}\bar{u}_{-\lambda}(r)\slashed{p}\gamma^\mu u_\lambda(r) \text{ and } \epsilon_0^\mu(p) = \frac{1}{m}\left(p^\mu - 2\frac{m^2}{2p\cdot r}r^\mu\right). \tag{21}$$

Eq. (20) and (21) both depend on a reference vector $r$, which defines the meaning of positive and negative helicity states for massive fermions and functions as a gauge choice for vector bosons. In both cases, we use

$$r = (1, -\vec{e}_p), \tag{22}$$

where $\vec{e}_p$ is a vector of unit size that points in the direction of $p$. It can then be shown that the spinors of eq. (20) correspond with the usual definition of helicity as the projection of spin along the direction of motion, and that the polarisation vectors of eq. (21) are respectively purely transverse and longitudinal.

The electroweak couplings are defined in terms of the weak mixing angle

$$c_w \equiv \cos\theta_w = \frac{m_w}{m_z} \quad s_w \equiv \sin\theta_w. \tag{23}$$

Table 1: Values of the EW coupling constants.

| | $v$ | | | $a$ | | | | $g_v$ |
|---|---|---|---|---|---|---|---|---|
| | $\gamma$ | $W$ | $Z$ | $\gamma$ | $W$ | $Z$ | $WW\gamma$ | $-e$ |
| $d$ | $-\frac{1}{3}e$ | $\frac{-e}{\sqrt{8}s_w}$ | $\frac{-e}{4s_wc_w}\left(1-\frac{4}{3}s_w^2\right)$ | $0$ | $\frac{-e}{\sqrt{8}s_w}$ | $\frac{-e}{4s_wc_w}$ | $WWZ$ | $-e\frac{c_w}{s_w}$ |
| $u$ | $\frac{2}{3}e$ | $\frac{-e}{\sqrt{8}s_w}$ | $\frac{e}{4s_wc_w}\left(1-\frac{8}{3}s_w^2\right)$ | $0$ | $\frac{-e}{\sqrt{8}s_w}$ | $\frac{e}{4s_wc_w}$ | | $g_h$ |
| $e$ | $-e$ | $\frac{-e}{\sqrt{8}s_w}$ | $\frac{-e}{4s_wc_w}\left(1-4s_w^2\right)$ | $0$ | $\frac{-e}{\sqrt{8}s_w}$ | $\frac{-e}{4s_wc_w}$ | $HWW$ | $-e\frac{m_w}{s_w}$ |
| $v$ | $0$ | $\frac{-e}{\sqrt{8}s_w}$ | $\frac{e}{4s_wc_w}$ | $0$ | $\frac{-e}{\sqrt{8}s_w}$ | $\frac{e}{4s_wc_w}$ | $HZZ$ | $-e\frac{m_w}{s_w}$ |

The coupling constants are defined in Table 1. There, $v$ and $a$ are the vectorial and axial couplings of $f\bar{f}V$ vertices, while $g_v$ and $g_h$ are the couplings of $VVV$ and $VVH$ vertices. The calculation of the antenna functions then proceeds by computing the relevant $1 \rightarrow 2$ collinear branching in the collinear limit, taking care to remove the gauge-dependent spurious terms associated with longitudinal polarisations as described in [15–17]. After taking the quasi-collinear limit, all reference vectors defined in eq. (22) become identical, and the branching processes can be expressed in Lorentz-invariant inner products that involve only the branching momenta and a common reference vector $r$. These are finally replaced by momentum-fraction variables that are defined in terms of the Vincia phase-space parameterisation.

## A.1 Final-Final Antennae

In the case of final-final antennae, the replacements

$$\frac{p_i \cdot r}{p_I \cdot r} \rightarrow x_i \equiv \frac{s_{ij} + s_{ik} + m_i^2}{m_{IK}^2} \quad \text{and} \quad \frac{p_j \cdot r}{p_I \cdot r} \rightarrow x_j \equiv \frac{s_{ij} + s_{jk} + m_j^2}{m_{IK}^2} \tag{24}$$

are made, where $m_{IK}^2 = (p_I + p_K)^2$ and $x_i$ and $x_j$ are collinear momentum fractions defined in terms of the Vincia phase-space variables. To further shorten notation, it is convenient to define

$$\tilde{m}_{ij}^2 = m_{ij}^2 - \frac{m_i^2}{x_i^2} - \frac{m_j^2}{x_j^2}. \tag{25}$$

Wherever appropriate, quark-flavour off-diagonal branchings are also included, weighted with the appropriate CKM matrix entry.

### A.1.1 Vector Boson Emission off (Anti)fermions

$$a_{f_\lambda \rightarrow f_\lambda V_\lambda}^{FF} = 2(v - \lambda a)^2 \frac{\tilde{m}_{ij}^2}{(m_{ij}^2 - m_I^2)^2} \frac{1}{x_j},$$

$$a_{f_\lambda \rightarrow f_\lambda V_{-\lambda}}^{FF} = 2(v - \lambda a)^2 \frac{\tilde{m}_{ij}^2}{(m_{ij}^2 - m_I^2)^2} \frac{x_i^2}{x_j},$$

$$a_{f_\lambda \rightarrow f_{-\lambda} V_\lambda}^{FF} = 2\frac{1}{(m_{ij}^2 - m_I^2)^2}\left((v - \lambda a)m_i\frac{1}{\sqrt{x_i}} - (v + \lambda a)m_I\sqrt{x_i}\right)^2,$$

$$a^{FF}_{f_\lambda \to f_\lambda V_0} = \frac{1}{(m^2_{ij} - m^2_I)^2}\left[(v - \lambda a)\left(\frac{m^2_I}{m_j}\sqrt{x_i} - \frac{m^2_i}{m_j}\frac{1}{\sqrt{x_i}} - 2m_j\frac{\sqrt{x_i}}{x_j}\right) + (v + \lambda a)\frac{m_I m_i}{m_j}\frac{x_j}{\sqrt{x_i}}\right]^2,$$

$$a^{FF}_{f_\lambda \to f_{-\lambda} V_0} = \frac{(m_I(v + \lambda a) - m_i(v - \lambda a))^2}{m^2_j}\frac{\tilde{m}^2_{ij}}{(m^2_{ij} - m^2_I)^2}x_j. \tag{26}$$

The antenna for vector boson emission off antifermions are identical with the exception of the exchange $(v \pm \lambda a) \leftrightarrow (v \mp \lambda a)$.

### A.1.2 Higgs Emission off (Anti)fermions

$$a^{FF}_{f_\lambda f_\lambda H} = \frac{e^2}{4s^2_w}\frac{m^4_i}{s^2_w}\frac{1}{(m^2_{ij} - m^2_I)^2}\left(\sqrt{x_i} + \frac{1}{\sqrt{x_i}}\right)^2,$$

$$a^{FF}_{f_\lambda f_{-\lambda} H} = \frac{e^2}{4s^2_w}\frac{m^2_i}{s^2_w}\frac{\tilde{m}^2_{ij}}{(m^2_{ij} - m^2_I)^2}x_j. \tag{27}$$

### A.1.3 Higgs Emission off Vector Bosons

$$a^{FF}_{V_\lambda \to V_\lambda H} = \frac{e^2}{s^2_w}\frac{m^4_v}{m^2_w}\frac{1}{(m^2_{ij} - m^2_I)^2},$$

$$a^{FF}_{V_\lambda \to V_0 H} = \frac{e^2}{2s^2_w}\frac{m^2_v}{m^2_w}\frac{\tilde{m}^2_{ij}}{(m^2_{ij} - m^2_I)^2}x_i x_j,$$

$$a^{FF}_{V_0 \to V_\lambda H} = \frac{e^2}{2s^2_w}\frac{m^2_v}{m^2_w}\frac{\tilde{m}^2_{ij}}{(m^2_{ij} - m^2_I)^2}\frac{x_j}{x_i},$$

$$a^{FF}_{V_0 \to V_0 H} = \frac{e^2}{4s^2_w}\frac{1}{m^2_w}\frac{1}{(m^2_{ij} - m^2_I)^2}\left(m^2_I - 2m^2_i\left(x_i + \frac{1}{x_i}\right)\right)^2. \tag{28}$$

### A.1.4 Vector Boson Splitting to Fermion-Antifermion

$$a^{FF}_{V_\lambda \to f_\lambda \bar{f}_{-\lambda}} = 2(v - \lambda a)^2\frac{\tilde{m}^2_{ij}}{(m^2_{ij} - m^2_I)^2}x^2_j,$$

$$a^{FF}_{V_\lambda \to f_{-\lambda} \bar{f}_\lambda} = 2(v + \lambda a)^2\frac{\tilde{m}^2_{ij}}{(m^2_{ij} - m^2_I)^2}x^2_i,$$

$$a^{FF}_{V_\lambda \to f_{-\lambda} \bar{f}_{-\lambda}} = 2\frac{1}{(m^2_{ij} - m^2_I)^2}\left((v + \lambda a)m_i\sqrt{\frac{x_j}{x_i}} + (v - \lambda a)m_j\sqrt{\frac{x_i}{x_j}}\right)^2,$$

$$a^{FF}_{V_0 \to f_\lambda \bar{f}_\lambda} = \frac{((v + \lambda a)m_i - (v - \lambda a)m_j)^2}{m^2_I}\frac{\tilde{m}^2_{ij}}{(m^2_{ij} - m^2_I)^2},$$

$$a^{FF}_{V_0 \to f_\lambda \bar{f}_{-\lambda}} = \frac{1}{(m^2_{ij} - m^2_I)^2}$$

$$\times \left[(v - \lambda a)\left(2m_I\sqrt{x_i x_j} - \frac{m^2_i}{m_I}\sqrt{\frac{x_j}{x_i}} - \frac{m^2_j}{m_I}\sqrt{\frac{x_i}{x_j}}\right) + (v + \lambda a)\frac{m_i m_j}{m}\frac{1}{\sqrt{x_i x_j}}\right]^2. \tag{29}$$

### A.1.5 Higgs Splitting to Fermion-Antifermion

$$
\begin{aligned}
a^{FF}_{H\mapsto f_\lambda \bar{f}_\lambda} &= \frac{e^2}{4s_w^2} \frac{m_i^2}{s_w^2} \frac{\tilde{m}_{ij}^2}{(m_{ij}^2 - m_I^2)^2}\,, \\
a^{FF}_{H\mapsto f_\lambda \bar{f}_{-\lambda}} &= \frac{e^2}{4s_w^2} \frac{m_i^4}{s_w^2} \frac{1}{(m_{ij}^2 - m_I^2)^2} \left( \sqrt{\frac{x_i}{x_j}} - \sqrt{\frac{x_j}{x_i}} \right)^2\,.
\end{aligned}
\tag{30}
$$

### A.1.6 Transverse Vector Boson Splitting to Two Vector Bosons

$$
\begin{aligned}
a^{FF}_{V_\lambda \mapsto V_\lambda V_\lambda} &= 2g_v^2 \frac{\tilde{m}_{ij}^2}{(m_{ij}^2 - m_I^2)^2} \frac{1}{x_i x_j}\,, \\
a^{FF}_{V_\lambda \mapsto V_\lambda V_{-\lambda}} &= 2g_v^2 \frac{\tilde{m}_{ij}^2}{(m_{ij}^2 - m_I^2)^2} \frac{x_i^3}{x_j}\,, \\
a^{FF}_{V_\lambda \mapsto V_{-\lambda} V_\lambda} &= 2g_v^2 \frac{\tilde{m}_{ij}^2}{(m_{ij}^2 - m_I^2)^2} \frac{x_j^3}{x_i}\,, \\
a^{FF}_{V_\lambda \mapsto V_\lambda V_0} &= g_v^2 \frac{1}{(m_{ij}^2 - m_I^2)^2} \frac{(m_I^2 - m_i^2 - \frac{1+x_i}{x_j} m_j^2)^2}{m_j^2}\,, \\
a^{FF}_{V_\lambda \mapsto V_0 V_\lambda} &= g_v^2 \frac{1}{(m_{ij}^2 - m_I^2)^2} \frac{(m_I^2 - m_j^2 - \frac{1+x_j}{x_i} m_i^2)^2}{m_i^2}\,, \\
a^{FF}_{V_\lambda \mapsto V_0 V_0} &= \frac{g_v^2}{2} \frac{(m_I^2 - m_i^2 - m_j^2)^2}{m_i^2 m_j^2} \frac{\tilde{m}_{ij}^2}{(m_{ij}^2 - m_I^2)^2} x_i x_j\,.
\end{aligned}
\tag{31}
$$

### A.1.7 Longitudinal Vector Boson Splitting to Two Vector Bosons

$$
\begin{aligned}
a^{FF}_{V_0 \mapsto V_\lambda V_{-\lambda}} &= g_v^2 \frac{1}{(m_{ij}^2 - m_I^2)^2} \frac{(m_I^2(1 - 2x_i) + m_i^2 - m_j^2)^2}{m_I^2}\,, \\
a^{FF}_{V_0 \mapsto V_\lambda V_0} &= \frac{g_v^2}{2} \frac{(m_I^2 - m_i^2 + m_j^2)^2}{m_I^2 m_j^2} \frac{\tilde{m}_{ij}^2}{(m_{ij}^2 - m_I^2)^2} \frac{x_j}{x_i}\,, \\
a^{FF}_{V_0 \mapsto V_0 V_\lambda} &= \frac{g_v^2}{2} \frac{(m_I^2 + m_i^2 - m_j^2)^2}{m_I^2 m_i^2} \frac{\tilde{m}_{ij}^2}{(m_{ij}^2 - m_I^2)^2} \frac{x_i}{x_j}\,, \\
a^{FF}_{V_0 \mapsto V_0 V_0} &= \frac{g_v^2}{4} \frac{1}{m_I^2 m_i^2 m_j^2} \frac{1}{x_i^2 x_j^2} \\
&\quad \times \Big[ m_I^4 x_i x_j (x_i - x_j) + 2m_I^2 (m_i^2 x_j^2 (1 + x_i) - m_j^2 x_i^2 (1 + x_j)) \\
&\quad\quad - (m_i^2 - m_j^2)(m_i^2 x_j (1 + x_j) + m_j^2 x_i (1 + x_i)) \Big]^2\,.
\end{aligned}
\tag{32}
$$

### A.1.8 Higgs Splitting to Two Vector Bosons

$$
\begin{aligned}
a^{FF}_{H \mapsto V_\lambda V_{-\lambda}} &= \frac{e^2}{s_w^2} \frac{m_\nu^4}{m_w^2} \frac{1}{(m_{ij}^2 - m_I^2)^2}, \\
a^{FF}_{H \mapsto V_\lambda V_0} &= \frac{e^2}{2s_w^2} \frac{m_\nu^2}{m_w^2} \frac{\tilde{m}_{ij}^2}{(m_{ij}^2 - m_I^2)^2} \frac{x_j}{x_i}, \\
a^{FF}_{H \mapsto V_0 V_\lambda} &= \frac{e^2}{2s_w^2} \frac{m_\nu^2}{m_w^2} \frac{\tilde{m}_{ij}^2}{(m_{ij}^2 - m_I^2)^2} \frac{x_i}{x_j}, \\
a^{FF}_{H \mapsto V_0 V_0} &= \frac{e^2}{4s_w^2} \frac{1}{m_w^2} \frac{1}{(m_{ij}^2 - m_I^2)^2} \left( m_I^2 - 2m_\nu^2 \left( \frac{1}{x_i x_j} - 1 \right) \right)^2.
\end{aligned}
\tag{33}
$$

## A.2 Initial-Initial Antennae

In the case of initial-initial antennae, the replacements

$$
\frac{p_A \cdot r}{p_a \cdot r} \to x_A \equiv \frac{s_{AB} + s_{aj}}{s_{ab}} \quad \text{and} \quad \frac{p_j \cdot r}{p_a \cdot r} \to x_j \equiv \frac{s_{aj} + s_{bj} - m_j^2}{s_{ab}}
\tag{34}
$$

are made. The antenna functions are defined in terms of the collinear momentum fractions eq. (34). To shorten notation, we define

$$
\tilde{q}_{aj}^2 = x_A m_a^2 - \frac{x_A}{x_j} m_j^2 - q_{aj}^2.
\tag{35}
$$

### A.2.1 Vector Boson Emission off (Anti)fermions

$$
\begin{aligned}
a^{II}_{f_\lambda \mapsto f_\lambda V_\lambda} &= 2(v - \lambda a)^2 \frac{\tilde{q}_{aj}^2}{(m_A^2 - q_{ai}^2)^2} \frac{1}{x_A} \frac{1}{x_j}, \\
a^{II}_{f_\lambda \mapsto f_\lambda V_{-\lambda}} &= 2(v - \lambda a)^2 \frac{\tilde{q}_{aj}^2}{(m_A^2 - q_{ai}^2)^2} \frac{x_A}{x_j}, \\
a^{II}_{f_\lambda \mapsto f_{-\lambda} V_\lambda} &= 2 \frac{1}{(m_A^2 - q_{ai}^2)^2} \left( (v - \lambda a) \frac{m_A}{\sqrt{x_A}} - (v + \lambda a) \sqrt{x_A} m_a \right)^2, \\
a^{II}_{f_\lambda \mapsto f_\lambda V_0} &= \frac{1}{(m_A^2 - q_{ai}^2)^2} \\
&\quad \times \left[ (v - \lambda a) \left( \frac{m_a^2}{m_j} \sqrt{x_A} - \frac{m_A^2}{m_j} \frac{1}{\sqrt{x_A}} - 2m_j \frac{\sqrt{x_A}}{x_j} \right) + (v + \lambda a) \frac{m_a m_A}{m_j} \frac{x_j}{\sqrt{x_A}} \right]^2, \\
a^{II}_{f_\lambda \mapsto f_{-\lambda} V_0} &= \frac{((v - \lambda a) m_A - (v + \lambda a) m_a)^2}{m_j^2} \frac{\tilde{q}_{aj}^2}{(m_A^2 - q_{ai}^2)^2} \frac{x_j}{x_A}.
\end{aligned}
\tag{36}
$$

The antenna for vector boson emission off antifermions are identical with the exception of the exchange $(v \pm \lambda a) \leftrightarrow (v \mp \lambda a)$.

### A.2.2 Higgs Emission off (Anti)fermions

$$a^{II}_{f_\lambda f_\lambda H} = \frac{e^2}{4s_w^2} \frac{m_a^4}{s_w^2} \frac{1}{(m_A^2 - q_{ai}^2)^2} \frac{1}{x_A} \left( \sqrt{x_A} + \frac{1}{\sqrt{x_A}} \right)^2 ,$$

$$a^{II}_{f_\lambda f_{-\lambda} H} = \frac{e^2}{4s_w^2} \frac{m_a^2}{s_w^2} \frac{\tilde{q}_{aj}^2}{(m_A^2 - q_{ai}^2)^2} \frac{1}{x_A} x_j . \tag{37}$$

## B Evolution Integrals

This section contains the trial evolution integrals used to sample branchings in the electroweak shower. We compute an invertible form for the choices of evolution variables given by eq. (9) and for the trial antennae given by eq. (15). In practice, the evolution integrals are sampled by finding a constant overestimate of the auxiliary variable $\zeta$ and applying a veto if the sampled point is outside phase space.

### B.1 Final-Final

The final-final phase-space factorisation is given by

$$d\Phi_{n+1} = d\Phi_n \times d\Phi^{\text{FF}}_{\text{ant}}, \tag{38}$$

where

$$d\Phi^{\text{FF}}_{\text{ant}} = \frac{1}{16\pi^2} f^{\text{FF}}_{\text{Källén}} \Theta(\Gamma_{ijk}) ds_{ij} ds_{jk} \frac{d\phi}{2\pi} . \tag{39}$$

In this expression,

$$f^{\text{FF}}_{\text{Källén}} = \frac{m_{IK}^2}{\sqrt{\lambda(m_{IK}^2, m_I^2, m_K^2)}} , \tag{40}$$

where

$$\lambda(a, b, c) = a^2 + b^2 + c^2 - 2(ab + ac + bc) \tag{41}$$

is the Källén function and

$$\Gamma_{ijk} = s_{ij} s_{jk} s_{ik} - s_{jk}^2 m_i^2 - s_{ik}^2 m_j^2 - s_{ij} m_k^2 + 4m_i^2 m_j^2 m_k^2 \tag{42}$$

is the three-body Gram determinant that express the boundaries of the radiative phase space. The trial evolution integral is

$$\mathcal{A}^{\text{FF}}(Q_1^2, Q_2^2) = \frac{\hat{\alpha}}{4\pi} f^{\text{FF}}_{\text{Källén}} \int_{Q_2^2}^{Q_1^2} dQ^2 d\zeta \, a_{\text{trial}}(Q^2, \zeta) |J(Q^2, \zeta)| , \tag{43}$$

where $|J(Q^2, \zeta)|$ is the Jacobian associated with the transformation $s_{ij}, s_{jk} \to Q^2, \zeta$. We introduce

$$\zeta_1 = \frac{s_{jk} + m_j^2}{m_{IK}^2} \text{ and } \zeta_2 = 1 - \frac{s_{jk} + m_j^2 + m_k^2}{m_{IK}^2} , \tag{44}$$

which both have phase-space boundaries

$$\zeta_\pm = \frac{1}{2} \left( 1 \pm \sqrt{1 - 4\frac{p_\perp^2}{m_{IK}^2 - m_I^2 - m_K^2}} \right) . \tag{45}$$

in the limit $m_i = m_j = 0$, and where we have used that for all electroweak branchings $m_k = m_K$. Since the massive phase space is contained in the massless one, the requirement of the positivity of the three-body Gram determinant may be incorporated with a veto. Transforming to transverse momentum and the above definitions of the auxiliary variable, we find

$$\mathcal{A}^{\text{FF}}(p_{\perp,1}^2, p_{\perp,2}^2) = \frac{\hat{\alpha}}{4\pi} f_{\text{Källén}}^{\text{FF}} \int_{p_{\perp,1}^2}^{p_{\perp,2}^2} \frac{dp_{\perp}^2}{p_{\perp}^2} \left[ c_1^{\text{FF}} d\zeta_1 + c_2^{\text{FF}} \frac{d\zeta_2}{\zeta_2} + c_3^{\text{FF}} \frac{d\zeta_1}{\zeta_1} \frac{\zeta_1}{x_j} + c_4^{\text{FF}} d\zeta_1 \frac{m_I^2}{p_{\perp}^2} \right]. \quad (46)$$

Note that the ratio $\zeta_1/x_j < 1$, and as such it may be incorporated by a local veto.

## B.2 Initial-Initial

The initial-initial phase-space factorisation is given by

$$d\Phi_{\text{ant}}^{\text{II}} = \frac{1}{16\pi^2} \Theta(\Gamma_{ajb}) ds_{aj} ds_{bj} \frac{d\phi}{2\pi} \frac{s_{AB}}{s_{ab}^2}, \quad (47)$$

where $\Gamma_{ajb}$ is defined by eq. (42). The trial evolution integral is

$$\mathcal{A}^{\text{II}}(Q_1^2, Q_2^2) = \frac{\hat{\alpha}}{4\pi} \hat{R}_f \int_{Q_2^2}^{Q_1^2} dQ^2 d\zeta a_{\text{trial}}^{II}(Q^2, \zeta) |J(Q^2, \zeta)| \frac{s_{AB}}{s_{ab}^2}, \quad (48)$$

where $|J(Q^2, \zeta)|$ is the Jacobian associated with the transformation $s_{aj}, s_{bj} \rightarrow Q^2, \zeta$. We introduce

$$\zeta = \frac{s_{bj} - m_j^2 {}'}{s_{ab}}, \quad (49)$$

which has phase-space limits

$$\zeta_{\pm} = \frac{1}{2s} \left( s - s_{AB} - m_j^2 \pm \sqrt{(s - s_{AB} - m_j^2)^2 - 4p_{\perp}^2 s} \right) \quad (50)$$

as a result of the requirement $s_{ab} < s$, where $s$ is the total hadronic invariant mass. The evolution integral is

$$\mathcal{A}^{\text{II}}(p_{\perp,1}^2, p_{\perp,2}^2) = \frac{\hat{\alpha}}{4\pi} \hat{R}_f \int_{p_{\perp,1}^2}^{p_{\perp,2}^2} \frac{dp_{\perp}^2}{p_{\perp}^2} \frac{d\zeta}{\zeta(1-\zeta)} \frac{d\phi}{2\pi} \frac{s_{bj} - m_j^2}{s_{ab} x_j}. \quad (51)$$

The factor $(s_{bj} - m_j^2)/s_{ab} x_j < 1$ is incorporated through a local veto.

# C Breit-Wigner Sampling and Partial Widths for Resonances

In this appendix, we describe the procedure used to select masses and decay channels for resonances. Upon creation of a heavy resonance by the electroweak shower, an invariant mass is sampled from the relativistic Breit-Wigner distribution

$$\text{BW}(m^2) \propto \frac{m_0 \Gamma(m)}{(m^2 - m_0^2)^2 + m_0^2 \Gamma^2(m)}. \quad (52)$$

The decay width is given by a sum over partial widths

$$\Gamma(m) = \sum_{\{ij\}} \Gamma^{ij}(m), \quad (53)$$

where the sum runs over all decay channels. Because the electroweak shower produces resonances with definite helicity states, the partial widths should also be computed as such. We make use of the variables

$$y_i = \frac{m_i^2}{m^2}, \ y_j = \frac{m_j^2}{m^2} \ \text{and} \ y_0 = \frac{m_0^2}{m^2}. \tag{54}$$

The partial widths are given by

$$\Gamma_h^{ij}(m) = \alpha(m^2)\left(1 + \frac{\alpha_s(m^2)}{\pi}\right)\frac{N_c}{8s_w^2}\frac{m^3}{m_w^2}y_i\left(1 - 4y_i\right)^{\frac{3}{2}},$$

$$\Gamma_t^{ij}(m) = \alpha(m^2)\left(1 - 2.5\frac{\alpha_s(m^2)}{\pi}\right)\frac{1}{4s_w^2}\frac{m^3}{m_w^2}$$
$$\times\left[(y_0 + y_i + 2y_j)(1 + y_i - y_j) - 4y_i\sqrt{y_0}\right]\sqrt{\lambda\left(1, y_i, y_j\right)},$$

$$\Gamma_{v_\perp}^{ij}(m) = \alpha(m^2)\left(1 + \frac{\alpha_s(m^2)}{\pi}\right)\frac{N_c}{3}m\sqrt{\lambda\left(1, y_i, y_j\right)}$$
$$\times\left[(v^2 + a^2)\left(1 - \left(y_i - y_j\right)^2\right) + 3(v^2 - a^2)\sqrt{y_i y_j}\right],$$

$$\Gamma_{v_L}^{ij}(m) = \alpha(m^2)\left(1 + \frac{\alpha_s(m^2)}{\pi}\right)\frac{N_c}{6}m\sqrt{\lambda\left(1, y_i, y_j\right)},$$
$$\times\left[(v^2 + a^2)\left(2 - 3\left(y_i + y_j\right) + \left(y_i - y_j\right)^2\right) + 6(v^2 - a^2)\sqrt{y_i y_j}\right], \tag{55}$$

where $N_c = 3$ for decays to quarks and $N_c = 1$ for decays to leptons. These widths include full mass corrections, as well as $\mathcal{O}(\alpha_s)$ corrections in accordance with Pythia [2]. Note the appearance of $y_0$ in the top width. It appears due to the cancellation of gauge-dependent terms associated with the scalar component of the longitudinal polarisation of the $W$ boson. The corresponding Goldstone boson couples to the top quark through the Yukawa coupling, which corresponds with the on-shell mass. On the other hand, the kinematic mass $m$ may be off-shell and dictates the running of the width. We point out that here, and in the electroweak antenna functions, one could in principle account for the running of the Yukawa couplings, but such effects are currently neglected.

Technically, sampling of the Breit-Wigner distribution with running width is achieved through rejection sampling using an overestimate distribution of the form

$$\widehat{BW}(m^2) = \frac{1}{n}\left[b_1\frac{m_0\Gamma_0}{(m^2 - m_0^2)^2 + b_2^2 m_0^2\Gamma_0^2} + b_3\theta(m^2 > b_4 m_0^2)\frac{m_0}{(m^2 - m_0^2)^{3/2}}\right], \tag{56}$$

where the second term is required to match the behaviour of the running width at high masses. The values of parameters $b_1$ through $b_4$ are optimised using a simple Monte Carlo procedure.

If the resonance survives the EW shower long enough to reach its offshellness scale, it is decayed by selecting a channel with relative probability

$$P_{\text{chan}}^{ij} = \frac{\Gamma^{ij}(m)}{\Gamma(m)}. \tag{57}$$

Due to its definite helicity, the angular distribution of the decay products is not uniform. A polar angle in the centre-of-mass frame is thus selected with probability proportional to the $1 \to 2$ matrix element summed over the final-state spins. Then, a helicity configuration is selected according to the individual spin channels, after which the decayed state may be constructed, boosted to the event frame, and inserted in the event.

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
