# Peer review of "Interleaved Resonance Decays and Electroweak Radiation in the Vincia Parton Shower"

_SciPost Physics, doi:SciPost Phys. 12, 101 (2022)_

## Round 1 · Referee Report · Anonymous · 2021-12-2

Report
Yes, after the authors address my comments/questions.
Author: Rob Verheyen on 2022-01-12 [id 2089]
(in reply to Report 1 on 2021-12-02)Thanks for the careful reading and comments. Here we attach a response with a list of changes, which have now been resubmitted.
Author: Rob Verheyen on 2022-01-12 [id 2090]
(in reply to Report 2 on 2021-12-06)Thanks for the careful reading and comments. Here we attach a response with a list of changes, which have now been resubmitted.
Attachment:
response_2.pdf

---

## Round 1 · Referee Report · Anonymous · 2021-12-6

The manuscipt "Interleaved Resonance Decays and Electroweak Radiation in the Vincia Parton Shower" details the implementation of a novel treatment of resonances decaying within an evolving parton shower, and further adding EW splittings for a full description of the evolution of a high-$Q^2$ system to a low-$Q^2$ system within the full Standard Model. The paper contains many novel and interesting ideas that should be published, given the authors have addressed a number of questions and issues.

1) On p. 3, where the authors are listing other parton shower formulations incorporating electroweak splitting functions. However, arXiv:1403.4788 and arXiv:2108.10817 (at the minimum) are missing. Please amend.

2) On p. 4, the authors mention the use of running widths for an improved phenomenological description of a propagator. However, running widths generally lead to gauge violations, in particular within the constrained parameters of the EW sector, in particular at distance from the resonance pole. Could the authors please add a brief discussion of this, since they mention running widths as a viable option also far away from the resonance.

   Similar comments are needed around eq. (3) in Sec. 2.1.

3) On p. 6, in the three definitions to define $Q^2_{\mathrm{RES}}$, two of them imply that resonances with their most likely invariant mass, i.e. the peak of their Breit-Wigner distribution or pole mass $m_0$ are assigned a $Q^2_{\mathrm{RES}}$ in the vicinity of zero, and thus certainly smaller than the parton shower infrared cut-off. In physical terms, even though the resonance still has its nominal width soft gluons or photons would still be able to resolve it. Could the authors please comment on the physical reasoning behind these choices.

   Further, since $Q^2_{\mathrm{RES}} < Q_{\mathrm{cut}}$, the following question presents itself: For resonance like the top where the physical width is close to typical values of the parton shower cut-off $Q_{\mathrm{cut}}$, this implies that a significant fraction of events reaches $Q_{\mathrm{cut}}$ in essentially the standard narrow-width approximation. Please comment.

4) Along the same lines, the authors comment on p. 8, end of Sec. 2.1, whether top quarks with off-shellnesses below $1\,\mathrm{GeV}$ should be allowed to hadronise. Given the shape of the Breit-Wigner distribution and a top quark width of about $1.3\,\mathrm{GeV}$, this actually applies to a very significant fraction of the events. Could I invite the authors therefore to expand on the consequences of this speculation a little further?

5) On p. 9, Sec. 3.2, the authors detail their evolution variables, etc., for the EW splitting functions. However, it seems the quantities are only given for FF and II antennae. Is there a reason why no EW IF antennae are used?

6) On p. 12, Sec. 3.6 the overlap veto is discussed. It seems to me this is a topic relevant to constructing parton shower histories out of $n$-body final states. Could the authors please clarify its importance in pure parton shower forward evolution?

7) On p. 14, Fig. 3, what do the visible bands around the curves signify? Please detail.

8) On p. 15, Fig. 4, there is a typo in the right figure, I believe it should be $m > m_0 + 3\Gamma_0$.

9) On p. 20, Sec. 20, when validation their EW parton shower implementation in terms of the produced EW Sudakov logarithms, have the authors tried to validate them for more relevant physical theories against the known expression of Denner and Pozzorini, hep-ph/0010201 and hep-ph/0110114? This would help to identify whether in addition to the leading logarithms any of the different classes of subleading logarithms can be reproduced. In particular, in how far the authors approximate handling of spin is able to reproduce the spin-dependent subleading logarithms. In essence, a more detailed analysis of the results of Fig. 12 to this extent would greatly benefit the reader to gauge the reliability of this implementation.

10) On p. 23, which features of the EW shower and the fully coherent QED shower prohibit their simultaneous use? Please comment.

---

## Round 2 · Referee Report · Anonymous (Referee 1) · 2022-1-25

The authors have provided a detailed response with respect to comments 1, 2 and 5 of the first report. However, some issues remain with the third and fourth comments:

3 I appreciate the author's enthusiasm in providing detailed computation time analysis for their implementations. This, however being interesting, is out of place and just a comment on the computational price of the added assets would have been enough. I will not dwell on this point any further, but the authors might want to clarify how exactly this time consumption analysis is generated and how such a large computational load ($\sim$10-20%) is justified. The authors could also improve their manuscript concerning the behaviour of their splittings especially with regards to their treatment of EW soft-singularities.

4 Figure 12 of the manuscript, although outlining the efficiency of the utilized veto algorithm, does not showcase its effectiveness in different scenarios. It would be particularly interesting to see other observables in this context, e.g. the transverse momentum distributions of these events. Another interesting observation could be a comparison between say, explicit $VV + 2j$ against $2j$ with parton shower. The latter would particularly show how much EW background is expected from pure QCD radiations in EW-sensitive phase space.

The above modifications should not be taxing for the authors and after being considered, I will recommend the publication of this paper in SciPost.

---

## Round 2 · Referee Report · Anonymous (Referee 2) · 2022-1-27

Report

The authors have sufficiently addressed all points raised in my report.

---

## Round 2 · Author Response

We are grateful to the reviewers for their careful reading of the manuscript and their detailed comments. We reply to them here point-by-point. They are also added as replies to the previous version in pdf format.

Referee 1:

  1. We agree that the enumerated list describing the interleaved algorithm could be improved. We have significantly expanded it and have tied the steps more explicitly to the illustration in figure 1 (now figure 2). To keep the list itself relatively concise, some of the content (such as the scale definitions under point 2) were moved up into the body text in the introduction to section 2, which also allowed to comment more extensively on those choices, cf our response to point 3 of referee 2.

  2. We have expanded the discussion on the resolution measure, and the relevance of the parameter R.
    In short, the exact form of the resolution measure is not of importance as long as it correctly separates the singular regions of phase space associated with each shower. For internal consistency, we have chosen to stay as close as possible to the more limited implementation in Pythia's default showers.

  3. We have added two figures showing the computation time penalties for enabling interleaved resonance decays (fig 3) and the EW shower (fig 15). Regarding the point of validation of the individual EW splitting kernels, these have been validated to give identical collinear limits against the branching kernels of 2002.09248, which were calculated separately. We have added a note of this in the text. Furthermore, the tests shown in, for instance, figure 10 and 11 (now figs 12 and 13) serve as an indirect validation, as the EW shower would not be able to produce the correct spectra without the correct splitting kernels. This is also reflected in the text.

  4. A check against a theoretical prediction is shown in figure 10 (now figure 12). We had struggled to format this figure, and accept that its message was not conveyed clearly. We have now revised it by emphasising the exact ME curve (which serves as theory baseline) by using a thick black line for that. We also dashed the lines that represent the individual components of the full Vincia result. Finally, to reduce number of curves, we also dropped the Pythia result, since that is anyway unrelated to the validation of the Vincia treatment as such. We hope that the figure now shows much more clearly that, if one does not enable the overlap veto, the sum of the QCD and the EW path overcount the exact matrix element by a large amount (left pane). On the other hand, with the veto enabled, the shower lines up quite well with the exact matrix element (right pane).
    We have tried to clarify this further in the text.

  5. This should have been the jet radius R of the anti-kT algorithm. We have adjusted the figure accordingly.

Referee 2: 1. We agree that both references should be included. arXiv:2108.10817 was released on the same day as this manuscript, but arXiv:1403.4788 should have been included in the first place, which we apologize for. We have added both.

  1. The point about running widths leading to potential gauge violations is of course completely correct. We would like to keep our remarks about it relatively brief, since this aspect is not by itself a main point of our paper. In the introduction on p4, we have therefore sought to change the relevant paragraph while trying not to lengthen it, as follows:

    ...as well as options for allowing partial widths (and hence relative branching fractions) to vary with $Q^2$. The latter allows, for example, to account for kinematic thresholds and effects of running couplings across a reasonable range around the pole, but cannot be pushed too far, especially in the electroweak sector where masses and couplings are not independent of each other.

One further point to clarify is that the masses of resonances produced by the EW shower are also distributed according to Breit-Wigner distributions with running widths. As is already explained in Appendix C, the gauge violations that lead to dangerous high-energy behaviour are taken care of in the same way as in the calculation of the EW antenna functions. Furthermore, Figure 2 (4 in the new version) shows that at high invariant mass, for the high-energy resonances that the EW shower tends to produce, the shower dominates over the Breit-Wigner distribution, meaning that the exact treatment of running width effects are of little consequence in that region. We have verified that this is true for all SM resonances. A note of this has been added to the resonance matching section.

  1. In Vincia, the default cutoff scale for final-state evolution is 0.75 GeV, so the given definitions of Q_res are typically above the shower cutoff, for top, Z, and W resonances in the SM (which all have widths larger than 1 GeV). This means that, when one of the dynamical scale choices is selected, most of these resonances will still decay before the cutoff is reached. (Although the Q_res$ distributions are peaked at zero, the integrated probabilities below the shower cutoff are still less than 50%). We added a new figure 1 to highlight this, which shows the surviving fraction of t, Z, and W resonances as functions of the shower scale, for the three different dynamic-scale options. The region of typical shower cutoff values (0.5 - 1 GeV) is also illustrated.

The reasoning for using a measure of offshellness is given in the introductory paragraph to section 2, specifically strong ordering of propagator virtualities, which dictates which diagrams are enhanced and which are suppressed. The default choice, eq.(4), explicitly represents the denominator structure of eq.(3); the alternatives are mainly provided to make it possible to study the sensitivity to variations on this choice. We have added the following two paragraphs to the beginning of section 2 to elaborate on this:

The desire to connect with the strong-ordering criterion in the rest
of the perturbative evolution, as the principle that should dictate the
leading amplitude structures, leads us to prefer a dynamical
scale choice for 
resonance decays, whereby resonances that are highly off shell will
persist over shorter intervals in the evolution than ones that are almost
on shell. We note that this has the consequence 
that the on-shell tail will be resolvable by soft photons or gluons,
albeit suppressed by the survival fraction. To illustrate this,
fig.~1 shows the survivial fractions (denoted $\Delta_R$)
as functions of 
evolution scale, for $t$, $Z$, and $W$ resonances, for three different
options for dynamical scale choices, all of which are roughly
motivated by the propagator structure:
\begin{eqnarray}
i)~~~Q_\mathrm{RES}^2 & = & (m - m_0)^2~, \\[2mm]
ii)~~~Q_\mathrm{RES}^2 & \stackrel{\mbox{\tiny default}}{\equiv}
&\left(\frac{m^2 - m_0^2}{m_0}\right)^2 > 0~, \label{eq:offsh} \\[2mm]
iii)~~~Q_\mathrm{RES}^2 & \equiv & |m^2 - m_0^2|~,
\end{eqnarray}
where $m_0$ is the pole mass and $m$ its BW-distributed
counterpart. 
Near resonance, options \emph{i)} and \emph{ii)}, illustrated in the
left and middle panes of fig.~1, are functionally almost
equivalent, differing mainly just by an overall factor 2, while for
option \emph{iii)}, illustrated in the rightmost pane, $m = m_0 \pm \Gamma/2$
translates to $Q^2 \sim m_0 \Gamma$, so that option is primarily
intended to give an upper bound on the effect that
interleaving could have.

Alternatively, our model also allows for using a fixed
scale, $Q_\mathrm{RES} \equiv \Gamma$, irrespective of offshellness. 
In that case, the resonance will not be resolved at all by any photons or
gluons with scales $Q < \Gamma$. We regard this as a good starting
point for the width dependence but have not selected it as our default
since the fixed-scale choice by itself does not automatically extend
strong ordering to the resonance propagators; this can only be
achieved by allowing the choice to be dynamical. Our default choice,
eq.~(\ref{eq:offsh}), is constructed to have a \emph{median} scale of
$\left< Q_\mathrm{RES} \right> = \Gamma$, while simultaneously
respecting strong ordering event by event.
This implies that soft quanta will be able to resolve the resonance
with a suppressed magnitude $\propto \Delta_R$, which acts as a form
factor.

We comment on the $Q_res < Q_cut$ issue under point 4.

  1. We hope for the referee's understanding that it is not the intent of this paper, which already presents a significant body of work, to also develop possible non-perturbative aspects of the tail of long-lived (coloured) resonances. In this work, that tail is left to be treated just as it would have been in the conventional non-interleaved framework. For the record, in principle, our view is that yes, top hadrons could be formed from (part of) that tail, but we would probably not simply identify Q_cut with the formation of fully formed top hadrons. Rather, one still has to consider a range of scales, between Lambda_QCD and Q_cut, with Lambda_QCD a more appropriate formation time for actual hadronic states. In the interval one would be dealing with top quarks that have started to build up a confining field, but which decay before it is fully formed. This would be a fun project but not one that we believe we could complete without significantly delaying the publication of the work we have already done. Rather than speculate, we believe we do as much as we can, for now, by pointing out that this aspect in principle exists, but leave it for future studies to consider it more carefully.

  2. IF branchings are indeed not included in the current implementation. The reason is that, contrary to QCD, no natural choice for recoiler selection as a result of colour ordering exists. One option to select recoilers is indeed presented in sec. 3.3, and a similar procedure was outlined in arXiv:1611.00788. For now, as we make no attempt to correctly describe coherent EW gauge boson emissions, and the number of included antennae is already very large, we made the choice to limit the shower to FF and II branchings, which is sufficient to include the relevant singular limits. We have added an explanation of this to the manuscript.

  3. We agree that this concept is relevant for the construction of parton shower histories in the context of merging. In fact, as described in arXiv:2003.00702, Vincia's QCD shower is now sectorized, which means that only a single shower history path is associated with any particular phase space point. The overlap veto procedure has a similar function, in that it sectorizes the QCD and EW showers. The EW shower is currently not sectorized by itself, but if it were, the overlap veto would ensure only a single shower history, through either the QCD or the EW shower, would be associated with a phase space point. We have added a note of this to the manuscript.

  4. The shaded bands are the statistical uncertainties. We added this to the caption of figure 3 (now figure 5), hoping that this provides enough clarity for the other figures.

  5. Correct, we have fixed this and appreciate the level of detail the reviewer has inspected the manuscript with.

  6. We thoroughly agree that an analysis of the logarithmic accuracy of the EW shower would be very interesting and highly desirable. The EW sector presents some unique features, such as the mentioned spin-dependent subleading logarithms, for which it would be very intersting to find out how the current spin treatment performs. However, the recent work of the PanScales collaboration, the Deductor shower and the work of for instance arXiv:2003.06400 have shown that the analysis of the logarithmic accuracy of a parton shower is not a straightforward task. For instance, a numerical analysis requires running the shower in extreme limits, for which Pythia and Vincia are not currently equipped. We consider such an analysis to be out of the scope of the current paper, and hope to be able to return to this topic in the form of a dedicated study in the future. We have added a brief paragraph at the end of the conclusion describing possible paths for future research.

  7. The main difference is that the QED shower does not treat particle helicities, while this is required for the EW shower. While not a major obstacle, the inclusion of spin-dependent antenna functions in the coherent QED shower is not entirely straightworward, for instance due to the possibility of spin flips of massive particles due to photon emissions. We added a brief note of this to the manuscript.

---

## Round 2 · List of Changes

Listed in the author comments.

---

## Editorial Decision

published